# Detergent-Free Isolation of Membrane Proteins and Strategies to Study Them in a Near-Native Membrane Environment

**DOI:** 10.3390/biom12081076

**Published:** 2022-08-04

**Authors:** Bankala Krishnarjuna, Ayyalusamy Ramamoorthy

**Affiliations:** Department of Chemistry and Biophysics, Biomedical Engineering, Macromolecular Science and Engineering, Michigan Neuroscience Institute, The University of Michigan, Ann Arbor, MI 48109-1055, USA

**Keywords:** detergent-free membrane protein isolation, ionic and non-ionic polymers, lipid-nanodisc, membrane protein stability and structure, NMR, cryoEM

## Abstract

Atomic-resolution structural studies of membrane-associated proteins and peptides in a membrane environment are important to fully understand their biological function and the roles played by them in the pathology of many diseases. However, the complexity of the cell membrane has severely limited the application of commonly used biophysical and biochemical techniques. Recent advancements in NMR spectroscopy and cryoEM approaches and the development of novel membrane mimetics have overcome some of the major challenges in this area. For example, the development of a variety of lipid-nanodiscs has enabled stable reconstitution and structural and functional studies of membrane proteins. In particular, the ability of synthetic amphipathic polymers to isolate membrane proteins directly from the cell membrane, along with the associated membrane components such as lipids, without the use of a detergent, has opened new avenues to study the structure and function of membrane proteins using a variety of biophysical and biological approaches. This review article is focused on covering the various polymers and approaches developed and their applications for the functional reconstitution and structural investigation of membrane proteins. The unique advantages and limitations of the use of synthetic polymers are also discussed.

## 1. Introduction

Membrane proteins account for about 25% of human protein-coding genes [1]. More astoundingly, they account for 60% of drug targets [2]. The dominance of membrane proteins as drug targets despite the relatively small percent of proteins they represent emphasizes the importance of determining their three-dimensional structures. However, the study of membrane proteins has long been known to be one of the most technically challenging tasks in structural biology. Their location within the amphipathic lipid bilayer of a cell and the hydrophobic core regions that stabilize them have challenged the effectiveness of biophysical techniques (such as X-ray crystallography, cryoelectron microscopy (cryoEM), and nuclear magnetic resonance (NMR) spectroscopy) commonly used to study cytosolic proteins. While X-ray crystallography studies have reported structures of many membrane proteins, the common use of detergents is not desirable as they destabilize the structure and inactivate the function of membrane proteins. Numerous studies have well demonstrated the need for a lipid membrane environment to natively fold and stabilize the structure and enable the function of membrane proteins. Therefore, there has been significant interest in the development of membrane mimetics to overcome these challenges. The use of bicelles did overcome some of the challenges, but the use of detergent in bicelles and their low stability are major limitations. On the other hand, the introduction of nanodiscs composed of near-native lipid membranes opened avenues for structural and functional studies on membrane proteins by using a variety of biophysical techniques. The introduction of amphipathic proteins, peptides, and synthetic polymers has expanded the scope of nanodiscs. In particular, the use of polymers to isolate membrane proteins, along with lipids and other membrane components, directly from the cell membrane without the use of detergents has become an attractive methodology for studying membrane proteins in their native environment. In this review article, we cover the recent developments in the direct isolation and functional reconstitution of membrane proteins for structural biology studies.

## 2. Traditional Approaches to Express, Purify and Reconstitute Membrane Proteins

Membrane protein isolation has been performed using detergents for over 50 years now. The standard protocol for doing so has used *Escherichia coli* (*E. coli*) as an expression system for expressing both prokaryotic and eukaryotic (heterologous; animals/plants) membrane proteins [3,4,5]. The maintenance and handling of *E. coli* cells are relatively easy, and the protein expression in *E. coli* is inexpensive, with high protein yields compared to the eukaryotic expression system. Various bacterial vectors and strains are available for cloning and protein expression, respectively [3]. pET vector with a T7 promotor and the BL21(DE3) *E. coli* strain are famous for producing proteins. Luria-Bertani or terrific-broth are two media regularly used for culturing bacteria. Plasmid (vector carrying a target gene) is incorporated into *E. coli* expression strain by transformation and grown in Luria-Bertani media until the culture optical density at 600 nm (OD_600_) reached ~0.6. In the case of terrific-broth, the culture can be grown until the O.D_600_ of 1.0–1.5 before inducing protein overexpression [6]. Then the target protein is overexpressed by adding isopropyl β- d-1-thiogalactopyranoside (IPTG). The protein expression levels vary between different bacterial strains. Additionally, the IPTG concentration, temperature, time duration after overexpression, and aeration conditions for bacterial growth affect the yield of protein expression yields and protein solubility. Hence, it is recommended to test small-scale expression in various *E. coli* strains to select the best strain before a large-scale protein production. After selecting a suitable strain, the expression can be tested at different IPTG concentrations, time points, and temperature conditions to achieve optimal protein yields. The protein expressed at these different conditions is analyzed by sodium dodecyl sulfate-polyacrylamide gel electrophoresis (SDS-PAGE). Based on the protein band intensity, a suitable expression strain, the optimal duration of expression time, temperature, and IPTG can be determined and optimized for a better yield. Another critical parameter to consider is aeration during culturing; this is extremely important when expressing proteins in deuterium-based media, which is commonly used in the production of deuterated proteins for high-resolution NMR studies. Aeration is efficient when the culture mixing is proper, which can be achieved by changing the rotation speed (rpm) in the shaking incubator.

The overexpression of some heterologous proteins in standard *E. coli* BL21(DE3) strain interferes with bacterial growth (toxicity) and may cause bacterial death. The toxicity to cells is caused by various factors: (a) Increased levels of target mRNA, (b) overload of protein machinery components and target protein, and (c) codon bias between different organisms that affect target protein expression. *E. coli* exhibits bias towards using certain codons over eukaryotes for the same amino acid. Codon optimization tools (GenScript) are developed to change/optimize heterologous gene sequences to overcome codon bias problems. Additionally, some heterologous proteins may form aggregates in the bacterial cytoplasm that can interfere with bacterial growth. The plasmids encoding heterologous membrane proteins are generally more toxic than those encoding heterologous soluble proteins [7]. Hence, it is not easy to obtain good yields for membrane proteins, especially for those proteins that exhibit intrinsic instability. Other *E. coli* strains, such as C41(DE3) and C43(DE3) strains, are successful in expressing proteins that are difficult to express in BL21(DE3) strain [7].

The expression of membrane proteins in *E. coli* is limited by many factors. For example, the expression of G-protein coupled receptors (GPCRs) is challenging as the proteins require post-translational modifications that cannot be achieved in prokaryotic expression systems. A few eukaryotic proteins, when they are expressed in *E. coli*, are prone to form inclusion bodies [5,8]; hence they require an in vitro refolding step to attain a correctly folded functional state. Eukaryotic expression systems are used to overcome these limitations. The commonly used eukaryotic expression systems are yeast (*Saccharomyces cerevisiae*, *S. sombe*, *Pichia pastoris*), insect cell lines (*Spodoptera frugiperda* [Sf9 or Sf21], and *Trichoplusia ni* [Hi5]), mammalian cell lines (human embryonic kidney cells [HEK293], baby hamster kidney cells [BHK-21], monkey kidney fibroblast cells [COS-7], Chinese hamster ovary cells [CHO]), *A. thaliana*, *N. benthamiana,* and drosophila photoreceptor cells [9,10]. An engineered *P. pastoris* strain [11] is also developed to express human membrane proteins that possess cholesterol-binding nonannular regions [12,13]. The protein expression in insect cell lines is promoted by baculovirus (*Autographica californica*) infection. Cell-free expression systems based on the extracts of *E.coli,* wheat germ, insect, and mammalian cells are also available for membrane protein production [14].

Table 1 lists different types of detergents used for membrane solubilization and the purification of membrane proteins [15]. The properties of detergents, such as solubilization in an aqueous environment (high critical micellar concentration (CMC)), lead to the formation of micellar structures that partially mimic the amphipathic cell membrane, therefore making detergents to be suitable for extraction and in vitro characterization of membrane proteins.

Figure 1 shows a schematic representation of membrane protein purification. Once the cells are grown, and the desired protein is overexpressed, they are lysed by sonication or freeze/thaw cycles or mechanical disruption (Dounce homogenization). Lysozyme is added in order to disrupt the bacterial cell membrane. All of the protein purification steps are performed at ~4 °C, and protease inhibitors are included to avoid any proteolysis of the target protein. Low temperatures are recommended to decrease proteolysis. The membrane fractions are separated from the soluble materials by centrifugation and solubilized in a buffer containing a detergent or mix of detergents (Table 1). Recombinant membrane proteins are generally expressed as fusion proteins containing an affinity tag (for example, 6His-tag or GST-tag) for purification [14]. Membrane proteins without an affinity tag can be purified using ion-exchange chromatography methods. The fractions from the affinity/ion-exchange purification step are concentrated and further purified using size-exclusion chromatography. The presence of the target protein and its purity at each step of purification can be identified using the SDS-PAGE method [16].

The purified membrane proteins are reconstituted in a desired membrane mimetic for biophysical and biochemical characterization. The most commonly used membrane mimetics are detergent micelles, lipid bicelles, in meso methods, amphipols, nanodiscs, and peptidiscs [14,17,18,19]. Each of these mimetics requires different preparation techniques and vary in their ability to function as membrane mimetics. The most advanced membrane protein reconstitute systems are lipid nanodiscs. The lipid nanodiscs are bilayered membrane mimicking systems surrounded by a belt made up of proteins (membrane scaffold protein (MSP)) or peptides, or synthetic polymers. Various membrane proteins purified using traditional methods but reconstituted into one of these lipid nanodisc reconstitution systems for high-resolution NMR-based structural studies are listed in Table 2 [20]. Readers are recommended to read the published review articles for the details on these membrane mimetics [21,22,23,24,25,26,27].

## 3. Challenges with the Purification and Reconstitution of Membrane Proteins

Detergent-based protein purification methods create a non-native environment because of the removal of native lipids associated with the transmembrane domains of a membrane protein. Removing native lipids can lead to protein instability/denaturation by aggregation, precipitation, and degradation. Additionally, membrane proteins are inherently unstable and have short half-lives; hence, each step of membrane protein preparation using detergents can be a hurdle, and the yield is often low [59,60].

Structural studies using biophysical approaches, such as solution and solid-state NMR techniques, require long experiment times (a few hours to several days) and protein concentration in the high µM to mM range. Hence, reconstitution systems that can keep the protein in a stable form at high concentrations for long periods (several days) are required for structural studies using multidimensional NMR techniques. Although detergent-based solubilization methods have contributed enormously to the characterization of membrane proteins, they have several limitations [61]. (a) Detergent monomers have a single hydrophobic acyl chain or two very short hydrocarbon chains, so the area occupied by the head group is typically larger than the area occupied by the hydrophobic chains. Therefore, they self-assemble to form spherical species. By contrast, lipids are nearly cylindrical and possess two hydrophobic tails, causing the area to be occupied by hydrophobic tails and the head group more similar, which renders the self-assembly of lipids to form a planar lipid bilayer. Micelles, for example, have detergent head groups and tails in a less ordered structure, making the core of micelles less shielded; this can pose a problem if the aqueous environment comes in contact with the core. (b) Since the hydrocarbon chain has considerable exposure to the aqueous environment (shielded only partially), the lateral pressure on the transmembrane domains of membrane proteins is decreased substantially, which creates an unwanted, highly dynamic environment in detergent micelles. (c) The curved surface of micelles can affect the tertiary structure of a membrane protein and protein–protein complexes. (d) The lack of a native-like hydrophobic core of the membrane can alter the topology of transmembrane domains of a membrane protein and alter the interactions between the transmembrane domains within a membrane protein or between proteins. (e) The rapid exchange of detergent molecules occurs in micelles with monomers in the aqueous medium. (f) Detergent molecules replace the native lipids associated with transmembrane regions of proteins. The overall effect of this replacement varies between proteins depending on how significantly lipids affect the protein structure. Therefore, the use of micelles can severely limit the structural and functional studies on protein–protein complexes, oligomers, ion channels, and fusogenic proteins. Some specific examples are mentioned below.

Membrane protein exposure to non-native environments can cause conformational changes that may result in low-resolution structures, leading to misinterpretation of a protein’s functional mechanism [62,63,64]. For example, Mjs2P crystallization in a detergent environment produced two different conformations, which caused confusion when selecting the physiologically relevant structure. The reconstitution of γ-secretase in A8-35 (an amphipol) produced a high-resolution structure, while a lower resolution structure was produced when a detergent was used [64]. In the case of the drug-metabolizing CYP450 enzyme, clear signs that the enzyme could oxidize even the detergent molecules were observed (nonionic Triton N-101) [65]. In some cases, even mild detergents can affect membrane protein structure and function [66]. For example, the cytochrome*b*_6_*f* complex is active when it forms a dimer [66]. The treatment with the mild nonionic detergent HG caused the removal of endogenous lipids and the subsequent dissociation of Rieske protein from the complex, leading to the lack of dimer formation and overall protein dysfunction [66]. The structure and drug (hexamethylene amiloride)-binding properties of the SARS-CoV-2 envelope protein transmembrane domain determined by reconstituting in lipid bilayers suggests that bilayer-bound protein conformation is more active and structural distortions caused by detergents are avoided [67]. These observations suggest that detergents are not the ideal systems for the characterization of membrane proteins.

GPCRs need to be studied in their native lipid environment as they are regulated by the juxtaposition of specific lipid compositions [68,69]. For example, phosphatidylglycerol and phosphatidylethanolamine modulate the active and inactive states of the human β2-adrenergic receptor (β2R), respectively [68]. Likewise, the oxytocin receptor showed cholesterol-mediated structural changes and orientation of two ligands bound to the receptor [70]. Class A GPCRs prefer to interact with phosphatidylinositol-bisphosphate (over phosphatidylserine), which stabilizes the functional G-protein-bound state of the receptor, thus influencing the downstream G-protein signaling [71]. Thus, lipid composition can affect the conformation, orientation, function, and dynamics of a transmembrane protein [72,73]. Additionally, the different lipid composition of cells from one organ to another and between different cell organelles further challenge the traditional reconstitution methods.

There is no universal detergent/lipid system that can be used to study various membrane proteins. The thermostable integral membrane protein pyrophosphatase from *Thermotoga maritima* is selectively active in a few detergents [74]. Likewise, the BK Ca^2+^–activated K^+^ channel showed enhanced unitary conductance in the presence of phosphatidylserine compared to that in a nonionic lipid environment [75]. Hence, detergent screening is needed to find a suitable detergent/lipid system for the solubilization and reconstitution of a specific membrane protein in its stable, functional state [74,76,77]. Non-conventional detergents are under development to overcome some of the limitations posed by conventional detergents. Calixarene-based detergents that contain calixarene aromatic rings have been used to solubilize and stabilize membrane protein/complexes [78]. MNG is used to solubilize and stabilize the functional form of muscarinic M_3_ acetylcholine receptor (M_3_AchR) and other membrane proteins [79]. The membrane proteins that showed aggregation and showed issues with their solubility and activity in DDM micelles behaved well in the presence of MNG [79].

As mentioned above, membrane proteins can be reconstituted in a detergent-free lipid environment. Amphipathic MSP-based [80,81,82] and peptide-based [83] nanodiscs are commonly used systems for studying membrane proteins in the desired lipid environment [84,85]. However, the MSP protein or peptide in the nanodisc belt can interfere with the membrane protein characterization by many biophysical techniques, including circular dichroism (CD) and ultraviolet-visible (UV) absorption spectroscopy techniques. The type of lipids associated with the transmembrane region/s is critical for the structure and activity of membrane proteins [86,87]. However, in this case, protein purification still depends on detergents that remove the native lipids [88]. The rate of detergent removal during membrane protein reconstitution may affect the functional state of the protein. Membrane proteins may aggregate if the detergent removal is fast. Unfortunately, in the large soluble domain-containing proteins, it has been a common practice to ignore the transmembrane domains and study only their soluble domains as they are much easier to handle. The soluble domains are relatively easy to crystallize, such as in the case of CYP450. However, recent studies have shown the importance of the transmembrane domains on the folding, stability, REDOX complex formation, and drug metabolism [87,89,90,91,92,93]. Despite nanodiscs being superior to detergent/lipid reconstitution systems, the membrane protein purification still rely on the detergents that disrupts the native lipid-protein interactions and protein stability. Therefore, detergent-free approaches are essential for membrane protein purification and reconstitution.

## 4. Detergent-Free Isolation of Membrane Proteins Using Amphipathic Polymers

The annular lipids surrounding the surface of transmembrane regions make non-specific hydrophobic contacts, whereas the non-annular lipids that are present in between the transmembrane domains make specific contacts with the protein; they are therefore crucial for maintaining protein conformation and function. KcsA, for example, functions only in the presence of anionic lipids [94]. The structural studies confirm that the non-annular binding site containing positively charged Arg64 and Arg89 residues interact with the negatively charged phosphatidylglycerol [95]. The native lipids and their hydrophobic chain lengths can influence the structure, stability, function, and dynamics of membrane proteins [96,97]. Therefore, preserving the native lipids associated with the transmembrane domains of membrane proteins is crucial. Detergents usage affects such native lipid–protein interactions, and, subsequently, the protein function is affected, as observed for the human equilibrative nucleoside transporter-1 (hENT1) [98]. hENT1, which plays a crucial role in the cellular uptake of various anti-cancer and antiviral nucleoside analogs, showed increased thermal stability under detergent-free conditions [69,98]. In the preparation of cytochrome *bc*_1_ complex, decreasing the exposure time to detergents rendered protein with improved activity and increased phospholipids in the structure [99]. To avoid the detrimental effects of detergents, amphipathic polymers that solubilize lipid aggregates but do not disrupt the lipid–protein interactions have been used to isolate membrane proteins directly from cell membranes. The amphipathic polymers used for membrane protein reconstitution and detergent-free isolation are listed in Table 3 and Figure 2.

### 4.1. Solubilization of Cell Membranes and Isolation of Membrane Proteins into Local Lipid Polymer-Nanodiscs

The polymer-based purification protocol for membrane proteins in detergent-free conditions is similar to that of the detergent-based protocol (Figure 1) [108,109,110]. Protease inhibitors should be included in all the purification steps as most of the target proteins are protease-sensitive. Following cell lysis, the membranes are collected by centrifugation and washed with a high-salt buffer (250 to 500 mM NaCl) to remove most of the soluble cell components. If the target protein is not a metalloprotein, EDTA can be included in the lysis buffer to remove metal ions in the cell lysate. The membranes are then centrifuged again and resuspended in buffer (~pH 7.4 or required pH) either manually (pipetting/vortex) or by sonication. The membranes are then solubilized by adding polymer at a 1:0.5 to 1:1.5 membrane-to-polymer (weight: weight) ratio. Various factors, including pH, temperature, metal ions, and polymer type, affect the efficacy of a given polymer in solubilizing cell membranes (Figure 3) [111]. It may not be necessary to add excess polymer to increase membrane solubilization [112]. High polymer concentration can make the protein solution very dense, block the purification column, and lower protein yields [112]. An excess of polymer might also interact and interfere with the target protein’s activity [113]. Solubilization can be performed either at room temperature [69,114] or 4 °C [5,55] for 1 h to several hours with gentle mixing [115]. It is generally recommended to perform solubilization at lower temperatures if the target has an intrinsic stability issue such as thermal denaturation, precipitation, and aggregation. Polymers can only solubilize 40–70% of cell membranes [114,116]. One rationale for the limited solubilization efficacy of a given polymer could be due to the interference caused by the various cellular components present in crude cell extracts. Polymers are shown to be more effective in solubilizing the lipid-rich regions as compared to the lipid-poor (protein-dense) regions in the membranes (Figure 4) [114,117,118,119]. Therefore, the protein yields from the low-lipid regions are low. The solubilization of low-lipid and protein-dense regions can be improved by externally adding lipids [119]. In some instances, the solubilization can be improved by increasing polymer concentration, increasing NaCl concentration (200–500 mM), or by manipulating the pH of the solubilization buffer [111,120]. NaCl at ~300 mM concentration is optimal to solubilize *E. coli* membranes [111]. Higher ionic strengths in buffers sometimes can cause protein aggregation, as reported for the human β_2_-adrenergic receptor (rhβ_2_AR) [112]. If the polymer has a high styrene-to-maleic acid ratio, a high salt concentration can induce the polymer to aggregate at near physiological pH (Figure 5). Optimal NaCl concentrations in buffers could help with nanodiscs formation by removing the charge-charge repulsions between charged lipids and charged polymer [111]. After solubilization, remove the insoluble membrane particles by low-speed (~3000 rpm) centrifugation for 30 min to 1 h. At this stage, the membrane lipids are solubilized by the polymer, and polymer-lipid-membrane protein complexes are formed (Figure 1). Similar to MSP, SMA co-polymers, when mixed with cell membrane/synthetic lipids, form membrane mimicking flat-bilayered disc-like structures called polymer–lipid nanoparticles/nanodiscs (Figure 6) [121,122,123,124,125]. The target membrane protein in soluble local lipid-nanodiscs (in the supernatant) is then purified by either affinity chromatography [110] or ion-exchange chromatography. The efficacy of SMA in solubilizing cell membranes is affected by fluidity, thickness, lateral pressure profile, and charge density of lipid bilayer [126].

As both the SMA-based polymer (carboxylate groups) and resin (Ni^2+^-NTA) are charged molecules, they interact and affect the binding of a His-tagged protein to the resin; subsequently, it results in decreased protein yields [55,110,127]. The free polymer can also affect the functional analysis of membrane proteins [128]. Such unnecessary polymer binding is prevalent when the solution has an excess of the free polymer during the solubilization/reconstitution step. Hence, the resin should be washed extensively with a high salt buffer to remove as many undesired proteins and free polymer as possible. A high concentration of NaCl (~300 mM) in the solubilization buffer can reduce non-specific polymer-resin interactions to some extent [110]. Alternative tags (for example, glutathione S-transferase (GST)) with the same net charge, which does not interact with polymer, would be helpful to minimize these problems and achieve higher protein yields. The fractions are analyzed by SDS-PAGE and further purified by size-exclusion chromatography. The sample to load onto the size-exclusion column should not contain any insoluble/precipitated content as that can block the SEC column.

**Figure 6 biomolecules-12-01076-f006:**
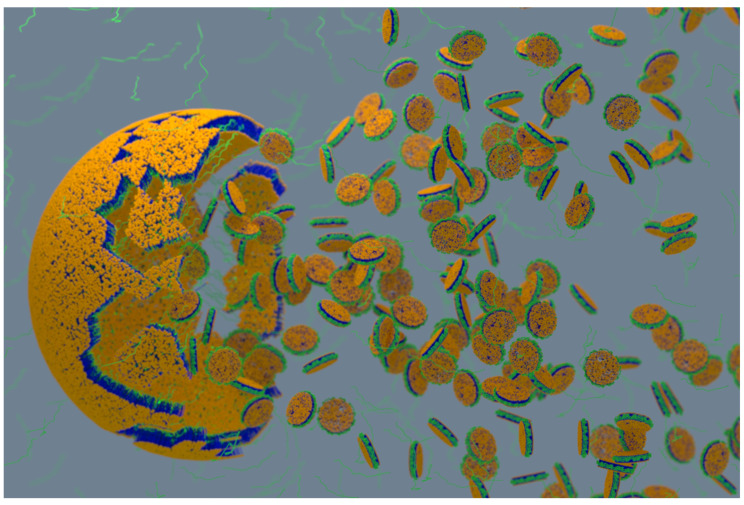
Schematic showing the formation of polymer-nanodiscs upon mixing synthetic lipids/membranes (liposomes; yellow/blue) with an amphipathic polymer (green). Polymer dissolves lipid aggregates and self-assemble to form discoidal nano-size particles called “polymer-nanodiscs”. The size of the nanodisc depends on the lipid:polymer ratio. The time course of dissolution and nanodiscs formation depends on the type of polymer and lipids used. The stability of nanodiscs against temperature, divalent metal ions (such as Ca^2+^ and Mg^2+^), and pH also depend on the type of polymer and lipids used. This Figure and caption are adopted from reference [129] with copyright permission.

The membrane protein/complexes can be solubilized directly from host tissues for functional studies [130]. However, the nanodiscs containing such protein/complexes may not be homogenous as they contain a range of different proteins from the membrane [130]. Therefore, it is challenging to separate unwanted proteins unless the target protein/complex is purified by protein-specific affinity chromatography.

### 4.2. Characterization of Local Lipid Polymer-Nanodiscs

SMA (~7–10 kDa) and DIBMA (~15 kDa) are the commonly used polymers to isolate membrane proteins in native lipids. Both polymers are amphipathic, negatively charged, and have detergent-like properties. Despite having “detergent-like” properties, these polymers neither denature membrane protein nor remove the lipids associated with the transmembrane domains. Thus, the membrane proteins and associated lipids can be isolated without the use of surfactants [117,131,132]. The native lipids associated with the purified membrane proteins can be identified/quantified by thin-layer chromatography (TLC) [113,117,127,130,133], mass spectroscopy (MS) [97,134,135,136,137], high-performance liquid chromatography (HPLC) [138], and ^31^P NMR spectroscopy [55] (Figure 7). The lipids from nanodiscs on TLC plates can be visualized using iodine or other staining methods, and the lipid types can be identified using synthetic lipids as a reference (Figure 7A–C). The native lipids can also be extracted using organic solvents (methanol/chloroform) for their analysis [117]. The identification of native phospholipids using ^31^P NMR spectroscopy does not require complex sample preparation protocols. The spectra of the isolated nanodiscs, however, exhibit broad spectral lines and render the analysis of lipids difficult. Therefore, the lipids in nanodiscs should be entirely dissolved by adding a detergent (cholate, for example) to separate the lipids from nanodiscs in the solution. The ^31^P NMR spectrum of such samples gives high-resolution spectral lines. Then, using the chemical shift values obtained from synthetic lipids (reference), the lipid types present in the nanodiscs sample are easily identified using its ^31^P NMR spectrum (Figure 7D) [55]. Although this simple 1D ^31^P NMR can only be used to identify ^31^P-containing phospholipids, other NMR experiments utilizing ^13^C and ^1^H nuclei can be used to identify any of the membrane components such as lipids, fatty acids, cholesterol, and polysaccharides. Gas-chromatography has been applied to identify the fatty acid composition of lipids in nanodiscs [117]. The native lipid analysis by MS is challenging due to the heterogenic nature of polymers in nanodiscs that generate a large range of different charge states. Laser-induced liquid bead ion desorption-MS (LILBID-MS) was applied to determine the oligomeric state of a membrane protein in native nanodiscs [137]. LILBID-MS can be applicable for identifying the annular lipids surrounding a membrane protein (lipid:protein ratio) in nanodiscs. An exchange protocol involving the replacement of polymers in nanodiscs with detergents/amphipols has been developed for native lipid analysis; however, the MS spectra showed broad peaks (Figure 7E,F) [134]. The low-resolution MS traces may be due to the incomplete polymer replacement in the samples. Further development is needed to overcome the low-resolution limitations posed by these MS methods [137]. The lipid composition of nanodiscs might vary from one cell to other cell types, and it is also affected by the growth conditions such as temperature and light [97,114]. The size of nanodiscs can be analyzed by dynamic light scattering (DLS) and transmission electron microscopy (TEM), while the protein–lipid complex in nanodiscs can be analyzed by cryo-electron microscopy (please see the section on protein structures by cryo-EM) and laser-induced liquid bead ion desorption-MS (LILBID-MS) [137].

### 4.3. Different Types of Membrane Proteins Isolated from Various Cell Membranes Using Amphipathic Polymers

The polymer-based membrane protein isolation strategy is the only method to isolate and study membrane proteins in their local lipid environment. Table 4 shows the list of membrane proteins isolated using the detergent-free isolation method. Although various SMA-based polymers are available, SMA (2:1) is the most commonly used polymer due to its efficacy in solubilizing cell membranes more efficiently than other variants. Initially, SMA polymers were used to reconstitute membrane proteins that were first purified using detergent-based isolation methods [121,131]. However, the membrane proteins purified using polymers in native lipids are more stable and active than those purified and reconstituted using detergent micelles [60,69,98,117,127,131,133,139,140,141,142]. For example, the full-length rhomboid protease of *Vibrio cholerae* (VcROM) and the human serotonin transporter (hSERT) expressed in *Pichia pastoris* and isolated using DIBMA are exceptionally stable and active. In contrast, they showed decreased activity and degradation/self-proteolysis when detergents were used in the purification protocols [127,142]. Likewise, the *Listeria* NDH-2a enzyme is difficult to purify as it is unstable when detergents are used. However, when the detergent-free direct-isolation method using SMA polymer was applied, a stable form was obtained for functional studies [60]. The activity of proteins prepared using detergent-free isolation methods in native lipid nanodiscs is similar to that in membranes [127] and functionally more active than those purified using detergents [143]. Thus, polymers are suitable for isolating and studying membrane proteins in the near native-lipid environment, especially G-protein coupled receptors [115,144,145] which are very challenging systems to study as they possess high intrinsic flexibility and lower stability in detergent micelles [69]. ABCG2, one of the ABC binding cassette proteins, has been directly isolated in its physiologically relevant oligomeric state (dimer) and used to investigate protein–drug interactions [145]. Smoothened (SMO) GPCR protein isolated in SMA-nanodiscs was active and enabled to measure dissociation constant (K_d_) for SMO ligands using NMR experiments [146]. *Drosophila* nicotinic acetylcholine receptor subunits were isolated into SMA-nanodiscs, and their native interactions with α-bungarotoxin (α-Btx; insecticidal peptide toxin) were studied [147]. P-glycoprotein (ABC transporter family) was isolated from MCF-7/ADR cells using SMA (SMALPs) and screened against 50 different natural products using surface plasmon resonance (SPR) spectroscopy [148]. This is the first study investigating membrane protein–ligand interactions in SMALPs by immobilizing SMALPs on an SPR biosensor chip [148]. Wild-type human dopamine receptor 1 (D1), one of the most challenging proteins to study, has been isolated from HEK293f cells using the detergent-free isolation method. The protein made through this method has been used to report the interaction of the neurotensin peptide with the D1 receptor in native lipids for the first time [115]. The adenosine 2A receptor (A2αR) has been isolated from *P. pastoris,* and its conformational changes in the presence of agonist and inverse agonist ligands have been investigated [149]. The method has also been applied to isolate different human potassium channels that are difficult to crystallize [150]. The isolated channels are highly stable, functional, less prone to form aggregates, and easier to make concentrated samples when compared to detergent-purified samples [150]. Membrane protein complexes containing photosystems I and II and light-harvesting complex have been isolated from spinach and pea thylakoids [151]. Yeast transmembrane sensor Wsc1 containing a large-soluble domain and a single transmembrane domain has been isolated, and its structural model was generated using various biophysical methods [152]. Yeast succinate dehydrogenase (Sdh4) has been isolated to study the regulation of coenzyme Q levels and how oxidative stress caused by polyunsaturated fatty acids is modulated by Cqd1 and Cqd2 proteins [153]. Different regions of bovine photoreceptors, including center (rhodopsin) and rim (ABCA4 and PRPH2/ROM1), were solubilized and isolated using the immune-affinity purification method [135]. The negative stain micrographs of directly isolated ABCA4 showed substantially increased density for the transmembrane domains due to native lipids. The lipid/fatty acid composition is entirely different between the center and rim regions. Such lipid–protein complexes are believed to be essential for the structure and function of region-specific proteins.

The solubilization efficacy of SMA polymer depends on the lipid composition, protein concentration, and degree of lipid order in the cell/organelle membranes [154,155]. In HELA cells, solubilization of the ER membrane is much faster than the solubilization of the plasma membrane [154]. Polymer solubilization efficacy also depends on the polymer-to-lipid ratio. In the case of a POPC/POPG lipid system, a lipid: polymer ratio of 1:1.25 is required to solubilize completely and form lipodiscs (~30 nm size) [124]. At lower lipid: polymer ratio (<1:1.25), the solution showed a powder chemical shift pattern in ^31^P NMR spectra, indicating the incomplete solubilization and anisotropic phase of the solution. Thus, the nanodiscs’ solubility and size can be modulated by changing the lipid to polymer ratio.

**Table 4 biomolecules-12-01076-t004:** List of membrane proteins isolated using polymers under detergent-free conditions.

Prokaryotic Protein	Eukaryotic Protein
SMA (Anionic)	
*Listeria* NDH-2a enzyme from the pathogenic bacteria *Listeria monocytogenes*, expressed in *E. coli* [60]	GPCR [human adenosine A2A receptor (A2AR)] expressed in *P. pastoris* and human embryonic kidney (HEK)293T cells [69]
*E. coli* rhomboid protease GlpG (with a six α-helix transmembrane domain) expressed in *E. coli* [97]	hENT1 expressed in Sf9 insect cells [98]
Bacterial divisome, single transmembrane protein ZipA [110]	Protein complex (CytcO along with Rcf subunits from *S. cerevisiae* subcellular membranes such as mitochondria) [113]
K^+^ importer A (KimA) from *Bacillus subtilis* and sodium–solute symporter protein (SSS) expressed in *E.coli* [137]	Wild-type human GPCR; dopamine receptor 1 (D1) expressed in HEK293f cells [115]
KcsA; it is a tetrameric potassium channel from *Streptomyces lividans* expressed in *E. coli* [117]	Human tetraspanins (four helical transmembrane domains) [116]
*E. coli* trimeric multidrug efflux transporter AcrB expressed in *E. coli* [156]	Protein complex from *Saccharomyces cerevisiae* subcellular membranes such as mitochondria [130]
Potassium importer KimA from *Bacillus subtilis* expressed in *E. coli* [157]	Human Pgp (P-glycoprotein; ABCB1) expressed in High Five (*Trichoplusia ni*) insect cells [140]
α-helical seven-transmembrane proton pump bacteriorhodopsin from *Haloquadratum walsbyi* (HwBR) expressed in *E. coli* [119]	Plant plasma membrane Na^+^/H^+^ antiporter SOS1 (Salt Overlay Sensitive 1) of *Arabidopsis thaliana* expressed in *P. pastoris* [158]
Holo-translocon (HTL)—a supercomplex of SecYEG–SecDF–YajC–YidC proteins [159]	RhD antigen from RBCs [160]
2×34 kDa cation diffusion facilitator protein from *Cupriavidus metallidurans* CH34 for proton-detected solid-state NMR [161]	A_2α_R and CGRP (GPCRs) receptors expressed in *P. pastoris* and Cos7 cells, respectively [162]
PglC and PglA of *Campylobacter jejuni* expressed in *E. coli* [163]	P-glycoprotein (ABC transporter family) isolated from MCF-7/ADR cells for small molecule screening by SPR [148]
The membrane tether protein ZipA and the ATP binding cassette (ABC) transporter BmrA are expressed in *E. coli* [162]	Human multidrug resistance protein 4/ABCC4 (MRP4/ABCC4) (expressed in *Sf*9 cell-lines) [141]
Penicillin-binding protein complex PBP2/PBP2a from *Staphylococcus aureus* [164]	Human adenosine 2α receptor (A_2α_R; GPCR) expressed in *Pichia pastoris* [149]
(a) BmrA from *Bacillus subtilis* (a homodimer; each monomer provides six transmembrane α-helices and a cytosolic nucleotide-binding domain, (b) LeuT from *Aquifex aeolicus* (an amino acid:sodium symporter; comprising 12 transmembrane helices; and (c) ZipA from *E. coli* (a single transmembrane helix with a large cytosolic domain) [165]	(a) Full-length pore-forming α-subunits hKCNH5 and hKCNQ1 of human neuronal and cardiac voltage-gated potassium (K_V_) channels, (b) the fusion protein comprising of an α-subunit hKCNQ1 and its regulatory transmembrane KCNE1 β-subunit (hKCNE1-hKCNQ1), expressed in mammalian COS-1 cells [150]
*E. coli* tyrosine kinase with two transmembrane helices for ^19^F-NMR [166]	Dynamic dhurrin metabolon from the microsomes of *Sorghum bicolor* [167], photosystem I light-harvesting chlorophyll II supercomplex from spinach [168]
SecYEG in complex with SecA from *E. coli* [169]	Human ATP binding cassette ABCG2 is expressed in HEK293T cells [145]
AcrB from *Salmonella typhimurium* [170]	Melatonin MT1R (GPCR) [144] expressed in *P. pastoris*
*E. coli* cytochrome *bo_3_*; expressed in *E. coli* [171]	Human tetraspanins CD81; expressed in *P. pastoris* [155]
Bacterial pLGIC; expressed in *E. coli* [172]	Slow anion channel 1 (SLAC1) from *Brachypodium distachyon*, its structure is determined using cryo-EM at 2.97 Å [173], expressed in *S. pombe*
Thermally stable rhodopsin from *Rubrobacter xylanophilus* rhodopsin (RxR) and an unstable one from *Halobacterium salinarum* sensory rhodopsin I [174]; expressed in *E. coli* cells BL21 (DE3)	Rhodopsin, ABCA4, and PRPH2/ROM1 from mice [135]
SARS-CoV-2 S glycoprotein expressed in 293T cells	Renal outer medullary potassium channels (ROMK) are expressed in *E. coli* [128]
Mycobacterial membrane protein large 3 (Mmpl3) from *Mycobacterium tuberculosis* [175]	Human β_2_-adrenergic receptor expressed in HEK293T cell line [112]
BAM-^MBP-76^EspP co-complex; expressed in *E. coli* BL21 (DE3) [136]	Trimeric photosystem I from the cyanobacterium *Thermosynechococcus elongates* [176].
Sav1866 (ABC transporter) from *Staphylococcus aureus*, expressed in *E. coli* BL21 [177]	Spinach and pea thylakoid membrane protein complexes [151]
KcsA expressed in *E. coli* BL21(λDE3), isolated suing SMA analogues [101]	Yeast transmembrane sensor Wsc1 [152]
	Succinate dehydrogenase (Sdh4) [153]
	*Arabidopsis thaliana* cytochrome-b5 expressed in *E. coli* [178]
	BAK protein from mitochondria of BAK KO U2OS cells [179]
	Smoothened (SMO) GPCR protein expressed Sf9 insect cells [146]
**DIBMA (anionic)**	
The membrane tether protein ZipA and the ATP Binding Cassette (ABC) transporter BmrA [162]; expressed in *E. coli*	GPCRs: A_2α_R and CGRP receptor; expressed in *P. pastoris* and Cos7 cells, respectively [162]
Bacterial OmpLA [132]; expressed in *E. coli* BL21 (DE3)	Human serotonin transporter (hSERT); expressed in *Pichia pastoris* [142]
*E. coli* rhomboid protease GlpG and *Vibrio cholerae* rhomboid protease [127]	
*E. coli* ZipA [180]	
**Glyco-DIBMA (anionic)**	
Voltage-gated K^+^ channel KvAP; expressed in *E. coli* [107]	
**SMA-QA (cationic) and SMA-EA (anionic)**	
	Rabbit cytochrome b5; expressed in *E. coli* [55]
**Polymethacrylate (PMA)**	
	Neurotensin type 1 receptor; expressed in Sf9 cells [181]
**Pentyl-inulin (Non-ionic)**	
	FBD domain of CYP450-reductase; expressed in *E.coli* [58]

### 4.4. High-Resolution Structure Determination of Membrane Proteins in Local Lipid Polymer-Nanodiscs Using Cryo-EM

Detergent-free membrane protein isolation has led to the high-resolution structural characterization of several membrane proteins in the local lipid-bilayer environments (Figure 8). One example is the multidrug efflux transporter AcrB, which is present in the inner cell membrane of Gram-negative bacteria. A sub-nm single-particle cryo-EM structure of *E. coli* AcrB transporter, prepared using the SMA-based direct-isolation method, was determined at a resolution of 8.8 Å [156]. Another research group, using cryo-EM reconstruction on the directly isolated AcrB (*E. coli*, K-12) using native cell membrane nanoparticles (NCMN) system, achieved a density map resolution of 3.2 Å [182]. Although there were many crystal structures available for AcrB, its mechanism of active transport remained unclear. This is because the structural information about the trimeric AcrB-native lipid interactions was missing from the available structures determined under non-native lipid/detergent environments [183]. The high-resolution structure of directly-isolated AcrB resolved 24 native lipid moieties in the central cavity and identified a number of hydrophobic and hydrogen bond interactions between the native lipids and the protein [182]. The AcrB D407A mutant was also produced using the detergent-free approach, and its structure was determined by cryo-EM. The structural differences between AcrB and its mutant are subtle, which is in contrast to the substantial structural differences between AcrB/AcrB mutants that are purified using the detergent-based protocol [182]. Recently, the first structure of the G288D mutated AcrB transporter (PDB id: 6Z12) was determined using a detergent-free protein isolation approach [170]. This mutation contributes to multidrug resistance in *Salmonella typhimurium.* Alternative complex III (AcIII) is another protein isolated using the detergent-free approach, and its structure is determined by cryo-EM at a resolution of 3.4 Å [184]. The lipid-anchored protein subunits through tri-acylated cysteine were structurally observed for the first time in a lipid bilayer. Additionally, many native lipids associated with the transmembrane domains were mapped [184]. The structure of KimA (KUP family) as a homodimer from *Bacillus subtilis* was determined using cryo-EM at 3.7 Å resolution and demonstrated that KimA functions as a K^+^/H^+^ symporter [157]. A low-resolution structure of the human ABC transporter, P-glycoprotein (ABCB1), was determined by cryo-EM at 35 Å [140]. The α-helical seven-transmembrane microbial rhodopsin (HwBr) was isolated using SMA polymer and transferred into the lipidic cubic phase (LCP) for in meso crystallization (SMA-LCP approach). HwBr structure at 2 Å was then determined by X-ray crystallography [119]. The structures of glycine receptors with full agonist glycine and partial agonists taurine and γ-amino butyric acid have been determined using cryo-EM [185]. The electron density for lipids surrounding transmembrane domains is observed at a low resolution. The exciting results are that the authors discovered previously unseen partial agonist (GABA/taurine)-bound closed conformational state of the receptor (with a pore size of 3 Å in diameter) along with open, desensitized, and expanded-open states. Additionally, in the presence of partial agonists, several long-lived states of the receptor are captured in cryo-EM that are short-lived and cannot be seen in the presence of Gly. Thus, high-resolution structures, together with electrophysiology experiments, provided insights into different conformational states of the glycine receptor that exist along the receptor reaction pathway. The high-resolution structure of proton-pumping cytochrome *bo*_3_ was determined in SMA-based nanodiscs [171]. In the structure, one equivalent of ubiquinone-8 in the substrate binding of protein, redox centres, water molecules in the channel, and several phospholipids are resolved. His98 from the subunit I (H98^I^) has two conformations as it forms an H-bond with ubiquinone-8 or with E14^I^. The authors propose that such dynamics of His98I facilitate H-transfer from UQ-8 to the periplasmic space upon the substrate oxidation. The trimeric structure of the plant slow anion channel 1 (SLAC1) from *Brachypodium distachyon* is determined using cryo-EM at 2.97 Å [173]; electron densities corresponding to single-chain lipid moieties (sphingolipids) are observed at the three-fold axis of the trimer and diglycerides at peripheral regions between the protomers [173]. SLAC1 is a crucial protein in abscisic acid-mediated stomatal closure. Due to high flexibility, the N- and C-terminal domains are obscured in the cryo-EM analysis. The structure of a bacterial pentameric ligand-gated ion channel (pLGIC) is determined at 2.5 Å resolution by cryo-EM [172]. The protein-associated native lipids such as phosphatidylglycerol and cardiolipin are identified. The structure of trimeric acid-sensing ion channel 1 from chicken (cASIC1) is determined at pH 7 (desensitized-state; PDB id: 6VTK) and pH 8 (resting-state; PDB id: 6VTL) using cryo-EM at ~2.8 and 3.7 Å, respectively [186]. ASICs belong to the epithelial sodium channel/degenerin superfamily of ion channel proteins expressed in central and peripheral nervous systems. The structures reveal the highly conserved His-Gly motif within the reentrant loop (17–40) at the intracellular, structurally disordered N-terminal region of cASIC1. The His-Gly motif is implicated in the gating and ion selectivity of the ASICs. Although the specific lipid types are unresolved, lipid-like densities surrounding the transmembrane domains are observed, suggesting the preservation of native lipid-protein interactions [186]. The high-resolution structural folding of *E. coli* O157:H7 outer membrane protein (OMP) (EspP; contains a β-seam [a 12-stranded β-barrel] and a canonical β-signal) in complex with BamA is determined by cryo-EM [136]. BamA is a heterooligomeric OMP that binds to β-signals and catalyzes the assembly of β-barrels (hence called β-barrel assembly machinery [BAM]) in the outer cell membrane. Native nanodiscs enabled the mapping of several intermediate stages of β-barrel folding by BamA. Additionally, substantial changes in the membrane surrounding the folding intermediates are found for the first time; and the membrane changes might have occurred to fold β-sheets toward BamA to form β-barrel. Multiple forms of lysophosphatidylethanolamine are identified along with PE, PG, and CL in the native nanodiscs.

The bacterial holo-translocon (HTL)—a supercomplex containing SecYEG–SecDF–YajC–YidC proteins, was isolated intact with *E. coli* native lipids, and the role of YidC in membrane protein insertion and assembly was proposed [159]. The directly-isolated membrane proteins with isotope labeling are suitable for a high-resolution solution and solid-state NMR studies [55,161]. The SMA-based detergent-free isolation is also applied to the plant proteins expressed in *Pichia pastoris* [158]. Both full-length SOS1 and truncated SOS1 are isolated, and the study found that the amino acid region between 460 and 480 is crucial for protein function [158]. It may be helpful to reinvestigate the structures of membrane proteins that are purified by detergent-based methods and artificially reconstituted in synthetic detergent lipids [187]. The metabolon (from *Sorghum bicolor*) containing all different enzymes that catalyze the formation of cyanogenic glucoside dhurrin was isolated. The purified SMA-solubilized nanodiscs were found to contain homo- and hetero-oligomers of CYP79A1, CYP71E1 enzymes as one of the abundant proteins in metabolon [167]. In contrast, no P450 enzymes are detected in the cholate-based metabolon extraction [167].

### 4.5. Other Applications of Detergent-Free Solubilization of Cell Membranes

The detergent-free isolation method, together with the LILBID-MS technique, has been applied to determine the oligomeric state of membrane proteins such as sodium–solute symporter protein (SSS) and a K^+^ importer A (KimA) [137]. Another study used the SMA-based approach to isolate and visualize rhesus(Rh)-D antigens in native erythrocyte membrane lipids [160]. The recognition of these antigens by their antibodies is highly conformation-dependent. Such protein detection by conformation-specific antibodies could be a sensitive diagnostic test for detecting membrane proteins/antigens. The conventional detergent-based approaches are not efficient for characterizing them. Therefore, the direct membrane protein isolation method is an excellent approach to screening different proteins in different cells linked to various infections/diseases and genetic disorders. Recently, prion multimers in native lipids were isolated directly from the brain tissues of Syrian hamsters and mice using various SMA co-polymers [138]. The native lipids associated with the isolated prion isolates are analyzed by HPLC [138]. The SMA-based membrane protein isolation approach, along with mutagenesis and bioorthogonal labeling methods, enables the single-molecule fluorescence-based labeling/analysis of membrane proteins in native lipids [163,188]. Likewise, the membrane proteins isolated using DIBMA-polymers are compatible with activity-based probes for functional studies and small molecule inhibitor screening [127].

### 4.6. Amphipathic Polymers Resistant to Metal Ions and Different pH Conditions

Functionalized SMA-based co-polymers such as positively-charged SMA-QA and negatively charged SMA-EA with molecular masses of ~2 kDa have also been developed to overcome the stability issues with SMA and DIBMA polymers [129,189]. SMA-QA is resistant to both extreme pH (~2.5 to 10) and high concentration (~200 mM) of Mg^2+^ and Ca^2+^ metal ions [189]. Likewise, SMA-EA is stable at a pH ~5 to pH 10; and also resistant to Ca^2+^ (20 mM) and Mg^2+^ (30 mM) metal ions [129]. These new co-polymers have been used for in vitro reconstitution of various cytochrome proteins. Both SMA-EA and SMA-QA have been successfully used to isolate cytochrome-b5 in *E. coli* lipid-nanodiscs (Figure 9A) [55]. ^15^N-labelled cytochrome-b5 prepared using the detergent-free isolation method generated a high-quality 2D TROSY HSQC spectrum (Figure 9B). Furthermore, the native *E. coli* lipids associated with the isolated cytochrome-b5 are analyzed using ^31^P NMR spectroscopy (Figure 7D). Thus, the directly isolated membrane proteins are suitable for high-resolution NMR-based structural studies.

Polymethacrylate (PMA) is another amphiphilic co-polymer suitable for preparing lipid-nanodiscs (Figure 2). PMA consists of hydrophobic butyl methacrylate and cationic methacroylcholine chloride. Since PMA contains no SMA moiety, it does not interfere with the spectroscopic characterization of proteins in nanodiscs. Recently, the PMA co-polymer was used to purify neurotensin type 1 receptor (NTRS1) to demonstrate its viability for membrane protein isolation [181]. The polymer isolated protein in nanodiscs showed increased stability and activity than that purified using detergents.

## 5. Limitations of Ionic Polymers

While the synthetic SMA and DIBMA polymers have been well studied and their applications are well realized, there are some limitations in using them to study membrane proteins. SMA is sensitive to the presence of divalent metal ions (such as Ca^2+^ and Mg^2+^) and extreme pH (pH < 6) conditions (Figure 3 and Figure 5). The negatively charged carboxylic groups in polymers make them less stable in the presence of metal ions and near-acidic conditions. Since the maleic acid groups become protonated at low pH, SMA can self-assemble into insoluble aggregates [120], thus limiting the use of SMA to study membrane proteins, particularly those that function in acidic environments [190,191]. The solubility/aggregation of the SMA at different pH conditions is affected by the styrene to the maleic acid ratio (Figure 5) [120]. SMA polymer with low styrene to a maleic acid ratio (1.4:1) is shown to be suitable for solubilizing membrane proteins at low pH (pH < 6) [120]. SMA is sensitive to metal ions, precipitating at Mg^2+^ concentrations in the low mM range [162,192], limiting its use to study metalloproteins. The styrene moiety in SMA-copolymer also limits protein characterization by some spectroscopic techniques, as seen in the FT-IR analysis of *E. coli* rhomboid GlpG (EcGlpG) [127]. The styrene moiety can also interfere with the lipid acyl chains and, subsequently, with the conformation of transmembrane protein domains [127]. DIBMA, by contrast, is resistant to sub-millimolar concentrations of Mg^2+^ and Ca^2+^ ions. The metal ions are shown to improve the DIBMA-based solubilization of *E. coli* membranes and increase the final protein yields [193]. Additionally, DIBMA has no aromatic styrene group and therefore is less likely to interfere with the lipid acyl chain order and protein structural analysis by UV-based spectroscopic methods [127,162].

DIBMA forms larger nanodiscs than SMA; therefore, DIBMA-based nanodiscs may accommodate other proteins along with the target protein, leading to decreased protein yields compared to that obtained using SMA-copolymer [162]. Additionally, DIBMA is less efficient in solubilizing some of the membrane proteins as compared to SMA, and the stability of DIBMA-based nanodiscs appears to be lower than that of SMA-based nanodiscs [162]. However, the larger size of DIBMA can be advantageous in accommodating membrane proteins with multiple transmembrane domains (or protein–protein complexes), thus can provide a flexible environment for the protein to be active, as seen in the case of GPCR rhodopsin [194]. In contrast, SMA- and cationic poly(styrene-*co*-maleimide) (SMI)-based nanodiscs prevented the conversion of light-activated rhodopsin Meta-I form to the fully active rhodopsin Meta-II form, suggesting that the smaller size nanodiscs and the styrene moiety of the polymers may be restricting protein dynamics or causing an unwanted conformational change. SMA-to-rhodopsin ratios were also shown to affect the kinetics of protein, where a low polymer-to-protein ratio has been shown to affect only protein kinetics, whereas, at a high polymer-to-protein ratio, the protein became functionally inactive [195]. Therefore, one should be knowledgeable and cautious when using synthetic polymers in protein functional/structural studies. Cationic SMI is also resistant to Mg^2+^ and Ca^2+^ ions up to 100 mM concentration and stable at pH 5 but is also as effective as SMA at neutral pH in solubilizing *E. coli* membranes. Additionally, the SMI polymer has been shown to be compatible with the functional reconstitution of human adenosine A_2A_ receptor (A_2A_R) and the human V_1a_ vasopressin receptor (V_1a_R) expressed in HEK 293T cells [192]. The partial-esterification of SMA polymers caused lower protein yields and increased sensitivity to divalent ions [196]. The functionalization of maleic acid units in SMA with alkoxy ethoxylates of increasing alkoxy chain length improved the solubilization efficacy of trimeric photosystem-I from the membranes of cyanobacterium *Thermosynechococcus elongates* [176]. Recently, the DIBMA with a molecular mass of 3–7 kDa has been shown to be as efficient as SMA in solubilizing lipid membranes (~75%). The low-molecular-weight DIBMA migrates well; thus, it does not cause any smearing on SDS-PAGE analysis [180].

Unlike SMA, PMA is not sensitive to Ca^2+^ and Mg^2+^ ions up to 2 mM and 10 mM concentrations, respectively [132,181,192]. However, PMA is shown to affect protein yields; this may be by competing for Ca^2+^ binding with M1 antibody, thus inhibiting FLAG-tagged protein to the resin. Another polymer, methylstilbene-*alt*-maleic acid (STMA) co-polymers, whose composition can be controlled precisely, are also developed for membrane protein studies [197]. Similar to SMA, STMA is sensitive to divalent metal ions; it precipitates at above ~2.5 mM concentration of Ca^2+^ ions. In contrast, the SMA-based zwitterionic polymers are shown to be resistant to aggregation at low pH and can be suitable for isolating membrane proteins [106,198].

Despite being resistant to metal ions and different pH conditions, the use of SMA-EA and SMA-QA, due to the high charge density, to study membrane proteins with different net charges at a given pH, especially those containing large soluble domains, is limited as the non-specific charge–charge coulombic interactions affect the protein structure [111,199] (Figure 10). For example, CYP450 2B4 with a large soluble domain has a net positive charge (+6.9) at physiological pH 7.4 (Figure 11). The CYP450 2B4 was functionally active and stable when reconstituted in the positively charged SMA-QA-based lipid-nanodiscs. In contrast, CYP450 2B4 was only partially functional in lipid-nanodiscs formed by negatively charged SMA-EA polymer. This was because of non-specific charge–charge coulombic interactions between the positively charged CYP450 2B4 soluble domain and the negatively charged SMA-EA polymer.

Similarly, cytochrome-b5, which contains a large soluble domain (net charge −8.8 at pH 7.4), was reconstituted in polymer-nanodiscs, and its conformation was studied by NMR spectroscopy (Figure 12). The NMR spectral lines of cytochrome-b5 reconstituted in SMA-QA lipid nanodiscs were significantly broadened, indicating the high-molecular-weight polymer–protein aggregates formed due to non-specific ionic interactions. Although the non-specific interactions can be removed by adding a higher concentration of NaCl (500 mM) to the protein-nanodiscs sample (Figure 12), such high NaCl concentrations may not be compatible with the functional analysis of various membrane proteins [199]. Additionally, the polymer’s charge limits the functional reconstitution of membrane protein complexes with opposite net charges at a given pH. For example, electron transfer experiments between CYP450 reductase and CYP450 cannot be performed as these two proteins possess opposite net charges at pH 7.4. Therefore, one must be careful to choose the correct polymer with the same charge as the net charge on the target membrane protein.

SMA-EA and SMA-QA have been used to isolate cytochrome-b5 from *E. coli* cells in native lipids [55]. However, the protein yield using SMA-QA is lower than that of SMA-EA. SDS-PAGE confirmed that most of the protein is in the insoluble membrane portion, suggesting that SMA-QA may form aggregates by interacting with anionic *E. coli* membrane lipids and cytochrome-b5 [55,199]. Thus, the first step before purifying any membrane protein is to know the net charge of the target protein at a given pH of a buffer being used for purification/reconstitution and select the polymer accordingly. The recently developed anionic glycol-DIBMA with a decreased charge density showed better membrane solubilization than highly-charged DIBMA [107] (Figure 13). Thus, the inefficient membrane solubilization by SMA-based polymers (Figure 14) was because of charge-charge repulsions with anionic lipids in a membrane. Therefore, amphipathic polymers with decreased charge densities might possess better membrane solubilization properties than polymers with high charge densities. Additionally, low-charge polymers may have less effect on the protein conformation due to decreased non-specific interactions with the oppositely charged membrane proteins.

Polymers are less efficient in solubilizing cell membranes compared to detergents, resulting in significantly lower protein yields than proteins extracted using detergents [200]. Furthermore, the membrane/lipid solubilization can be affected by the chemical structure of polymers, the styrene-to-maleic acid ratio in SMA-based polymers, polymer length, and lipid composition of the cell membrane [128,180,200]. Furthermore, the lipids without and with membrane proteins can have different phase behavior; thus, a particular polymer can efficiently solubilize pure synthetic lipids but not the crude cell membranes [201]. The capability of solubilizing lipid membranes indicates that the polymers can interact and destabilize the cell membrane, which can generate false-positive results in the functional assays [128]. Some of the limitations, for example, the non-specific electrostatic interactions of polymer with the soluble domains of membrane proteins, can be overcome using non-ionic polymers.

**Figure 14 biomolecules-12-01076-f014:**
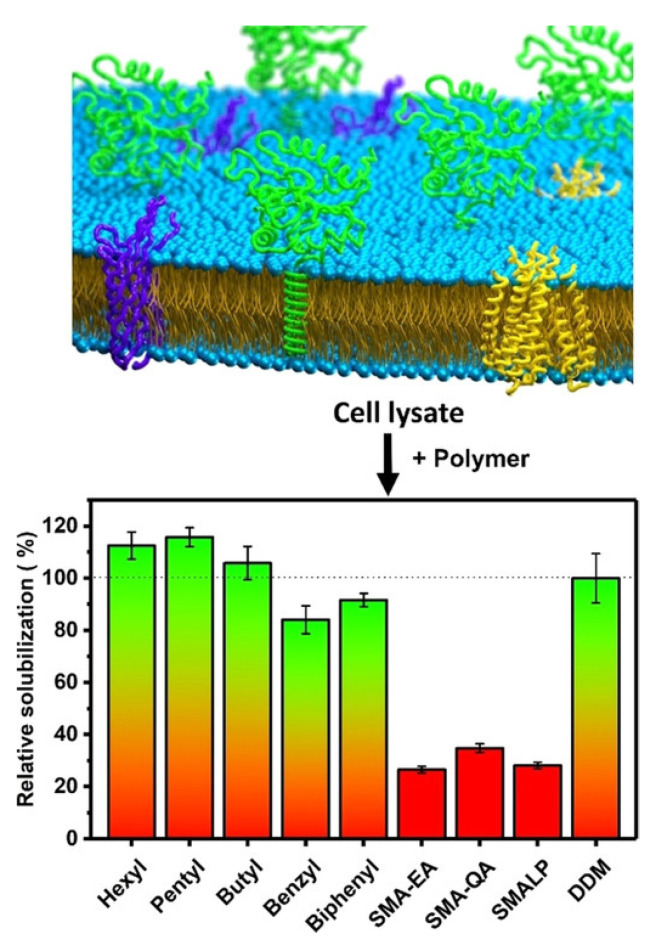
*E. coli* membrane solubilization by inulin-based non-ionic polymers possessing different types of hydrophobic functional groups (see Figure 2 for the chemical structure). The solubilization efficacy of inulin-based polymers is compared with that of SMA-based polymers and DDM. This Figure and caption are adopted from reference [202] with copyright permission.

## 6. Non-Ionic Polymers

Non-ionic inulin-based polymers were recently synthesized and have been shown to form nanodiscs (Figure 2) [202]. The polymers were synthesized by hydrophobic functionalization of inulin extracted from chicory root. The reaction involves the addition of a hydrophobic functional group to inulin. The identity of this group can vary and results in different lipid-solubilizing efficacy, as shown in Figure 14. The most efficient addition was found to be pentyl-inulin (i.e., pentane groups added to the inulin). The inulin-based polymers are shown to solubilize synthetic lipids (DMPC/DMPG) even in the presence of 100 mM concentration of divalent cations such as Ca^2+^ and Mg^2+^ and at a range of pH conditions (pH 2–8.5). Due to the non-ionic nature of the polymer, any non-specific electrostatic interactions with the target membrane protein are removed. Hence, in contrast to ionic polymers (SMA), non-ionic inulin polymers are compatible with the functional reconstitution of differently charged membrane proteins and complexes, as shown for a redox-complex containing oppositely charged CYP450 proteins [55,199,203]. Furthermore, non-ionic polymers are more efficient in solubilizing *E. coli* membranes than ionic SMA-based polymers, as summarized in Figure 14, and do not require high-salt concentrations and excess polymer [202]. Therefore, inulin polymers are a great new addition to the existing polymers for membrane protein studies. We have recently demonstrated the use of pentyl-inulin to isolate the FBD domain of CYP450-reductase from *E. coli* membranes and successfully demonstrate the feasibility of high-resolution NMR studies on the inulin-based nanodiscs containing FBD and native lipids [58]. The size of the nanodiscs was estimated by DLS and TEM experiments, and the lipids in the nanodiscs were identified by ^31^P NMR experiments. The isolated FBD in native *E. coli* lipid-nanodiscs was found to be structurally more homogeneous than that extracted using detergents, as shown by high-resolution NMR studies (Figure 15) [51,58]. These results demonstrate the suitability of inulin-based polymer-nanodiscs to isolate and reconstitute charged membrane proteins. Due to its non-ionic nature, pentyl-inulin is also compatible with protein purification using ion-exchange resins. Furthermore, non-ionic polymers do not interfere with the protein analysis by UV spectroscopy and SDS-PAGE (Figure 16) [58].

Recently, the saposin-lipoprotein (Salipro) approach, where adding a significantly low concentration of a mild detergent digitonin shown to increase the fluidity of membranes at 4 °C while keeping protein-associated native lipids intact [204]. Such an addition of a low concentration of mild detergents to the polymer-based isolation of very low-yielding membrane proteins may help to improve the membrane solubilization and protein yields while keeping the lipids intact. In general, membrane proteins are expressed in heterologous expression systems, so even when they are isolated under detergent-free conditions, the lipids associated with transmembrane domains are not fully-native [155]. For example, if human GPCRs are expressed in *E. coli* or yeast strains, then the isolated proteins are associated with the lipids of that particular organism. Non-native lipids might affect the membrane protein properties, as seen in the case of Ca^2+^-ATPase, where the protein showed optimal activity in C18 lipids but decreased activity in short or longer chain lipids [205]. To exactly mimic the native lipid composition, the desired lipids can be added after affinity purification and before the SEC purification steps of a polymer-based direct isolation protocol. Nanodiscs prepared in this manner might be relevant for investigating membrane protein interactions with their soluble protein/peptide ligands in a near-native lipid environment. EgK5, a plant defensin peptide, enters the plasma membrane of mammalian cells, modulates the function of voltage-gated K_V_1.3 channels, and suppresses the antigen-triggered proliferation of autoreactive T cells in autoimmune disease [206]. However, the structural mechanism of EgK5-mediated K_V_1.3 function within the membrane and its consequences, such as the depletion of phosphatidylinositol 4,5-bisphosphate (PIP_2_) in the cell membrane is unknown. Direct isolation and reconstitution of K_V_1.3 in native lipid-nanodiscs would help to explore such a novel mechanism of channel inhibition by peptide inhibitors [206,207,208,209,210,211,212,213] and can have broader implications for similar systems in the field. Merozoite surface protein 2 (MSP2) of *Plasmodium falciparum* is a GPI-anchored membrane protein and is a promising vaccine candidate [214,215,216,217]. However, the clinical trials have been carried out using the protein construct (intrinsically disordered protein with many sequence repeats) lacking the GPI-anchor. The structure of GPI-anchored MSP2 and the immunogenicity of membrane-anchored native protein are unknown; it is challenging to isolate native MSP2 directly from the parasite. The detergent-free membrane protein isolation method using amphipathic polymers in combination with the MSP2-specific antibody-mediated affinity purification can be applied to isolate GPI-anchored MSP2 in native-lipid nanodiscs for functional and structural studies. Thus polymer-based detergent-free membrane protein isolation method may be used to study a range of challenging membrane protein systems to understand various unknown biological problems.

## 7. Conclusions and Future Scope

The lipid membrane plays a vital role in the folding, structure, and dynamics of membrane proteins [218]. The introduction of lipid nanodiscs technology has expanded the scope of membrane mimetics and also enabled high-resolution structural and functional studies on different types of membrane proteins. The development of different types of nanodisc-forming amphipathic molecules such as scaffold protein, peptides, and synthetic polymers has overcome the many limitations of nanodisc technology. The ability to prepare nanodiscs of various sizes (from ~6 nm to ~60 nm diameter), lipid composition, and the charge of the belt have dramatically enhanced the value of the nanodiscs technology. In addition, the recent introduction of novel nanodiscs-forming amphipathic polymers exhibiting tolerance to divalent metal ions (Ca^2+^ and Mg^2+^) and pH variation would further expand the scope and applications of nanodiscs. The demonstration of detergent-free isolation and functional reconstitution of membrane proteins along with native lipids (and other membrane components) is remarkable, and the number of studies utilizing this approach for structural and functional studies of membrane proteins is naturally increasing. The ongoing developments for the production of charge-free nanodiscs-forming, the creation of nanodiscs composed of different types of lipid domains, and functional reconstitution of protein–protein complexes would result in optimized nanodiscs technology for high-throughput structural and functional studies of membrane proteins. Although most of the direct isolation of membrane protein studies used bacteria, the use of mammalian and other cell types and other organisms such as viruses, insects, and fungi will become common in the near future. It would be exciting to explore the use of the nanodiscs technology for other membrane-related studies as well [67,219,220,221,222,223,224,225,226,227,228,229,230,231].

## Figures and Tables

**Figure 1 biomolecules-12-01076-f001:**
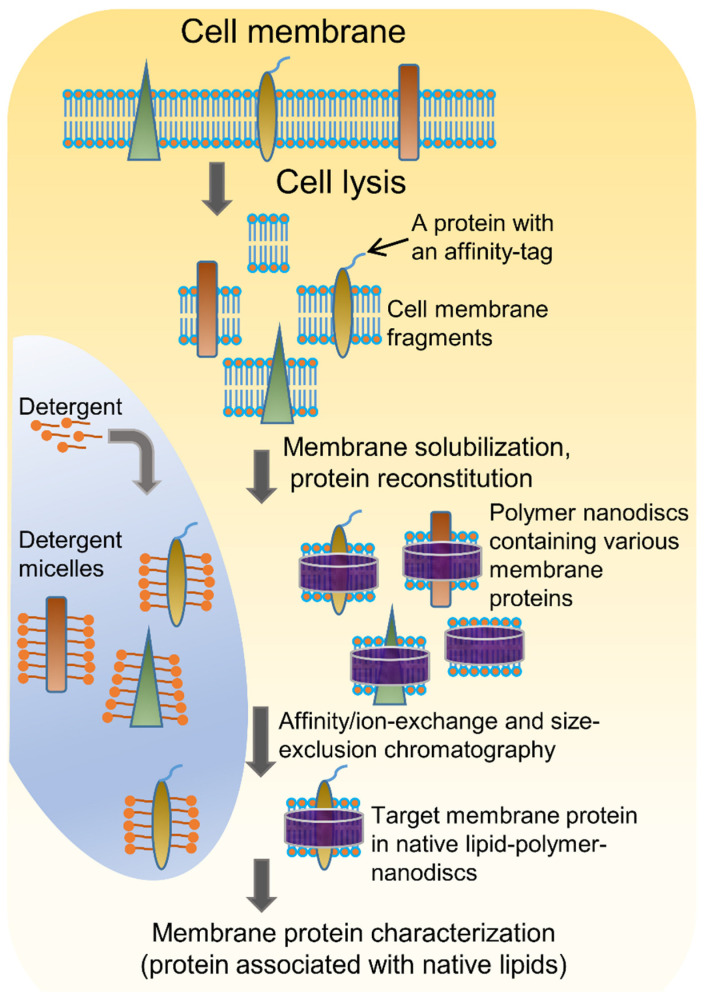
Schematic representation of membrane protein purification using the traditional detergent-based approach and the detergent-free polymer-based approach. The steps include protein expression, cell lysis, and purification. Unlike the detergent-based approach, in the detergent-free polymer-based method, a synthetic polymer is added to cell lysates to dissolve them to form polymer–membrane complexes from which nanodiscs containing the desired protein are isolated. In the nanodisc-membrane, the lipids are orderly packed, facing the hydrophobic tails inward and the hydrophilic head groups exposed to an aqueous environment.

**Figure 2 biomolecules-12-01076-f002:**
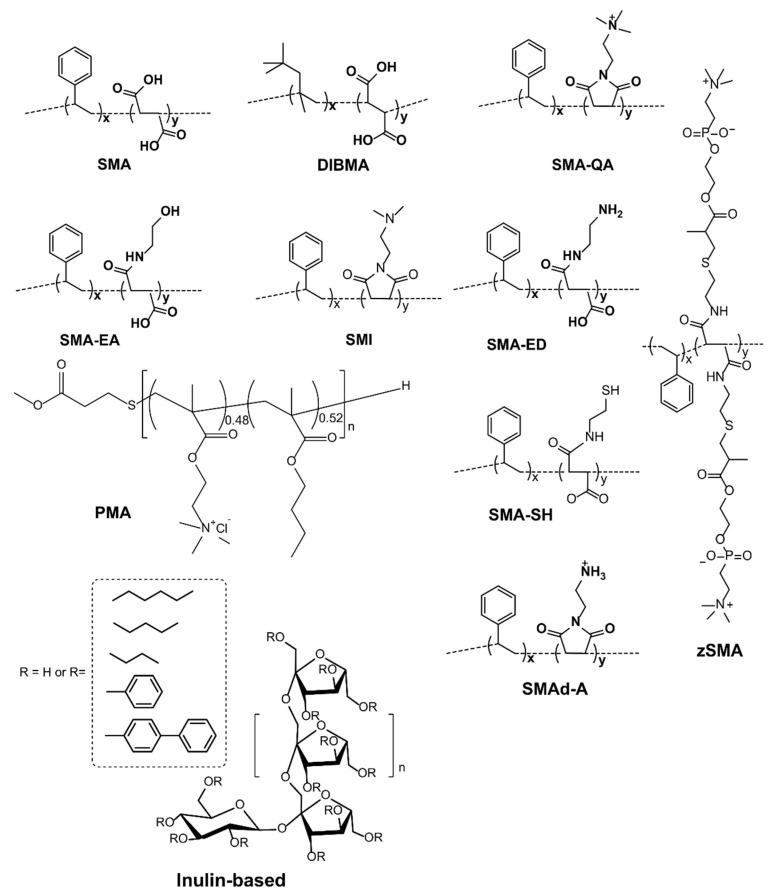
**Chemical structures of nanodiscs-forming synthetic amphipathic polymers.** These polymers have been developed and successfully shown to dissolve lipid-protein aggregates and form nanodiscs for reconstitution, detergent-free isolation, and characterization of membrane proteins in a near-native membrane environment. While research in this area continues to develop novel nanodisc-forming molecules (such as amphipathic polymers and peptides), these already reported polymers (including cationic, anionic, zwitterionic, and non-ionic) render studies on most (if not all) membrane proteins. An additional list of reported polymers can be found in the literature ([100,101,102,103,104,105,106,107]) and on the SMALP website (https://www.smalp.net/polymers.html, accessed on July 25 2022). The polymer structures were generated using ChemDraw [19.1.1.21].

**Figure 3 biomolecules-12-01076-f003:**
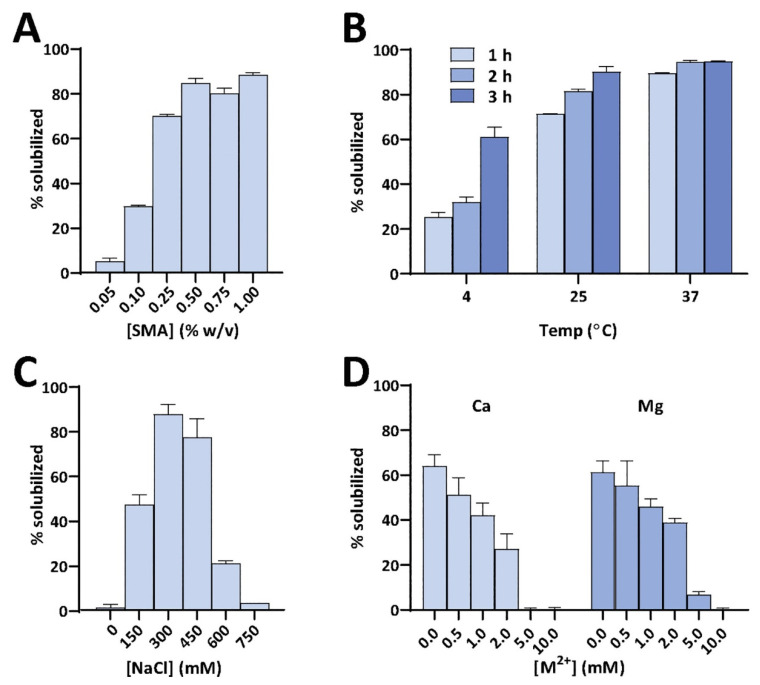
Influence of various environmental parameters on SMA-based solubilization of *E. coli* membranes expressing KcsA in 10 mM Tris buffer at pH 8. Other parameters were varied, as indicated in the figure, with standard conditions being: 0.25% (*w*/*v*) SMA, 2 h incubation at 25 °C in 300 mM NaCl, and 15 mM KCl. (**A**) Influence of SMA concentration. The amount of membrane material was kept constant, and SMA was added at different final concentrations in the range of 0.05–1% (*w*/*v*). (**B**) Influence of incubation time and temperature. (**C**) Influence of salt concentration. Different amounts of NaCl were added at a constant ratio of NaCl/KCl of 20. The sample devoid of NaCl contained 5 mM KCl to ensure the structural stability of the KcsA tetramer. (**D**) Influence of divalent cations (M^2+^). CaCl_2_ or MgCl_2_ was used at a concentration of 0–10 mM; all samples contained 15 mM KCl in Tris-HCl 50 mM, pH 8. Data are averages of 2 independent samples, with the error margin indicating the difference in solubilization between both samples. Overall, increasing SMA concentration, temperature (~25 to 37 °C), incubation time, and salt concentration (~300 to 450 mM) are shown to enhance the solubilization yield of KcsA. pH is also shown strongly influence efficiency of SMA, with maximum efficiency reached for pH 8 or 8.5. This Figure and caption are adapted with permission from reference [111].

**Figure 4 biomolecules-12-01076-f004:**
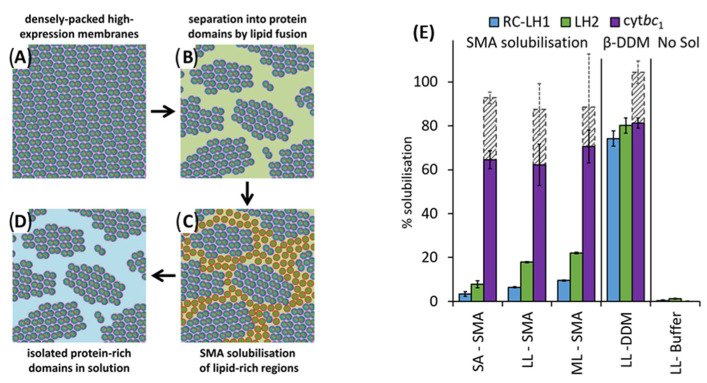
**Model for the formation of membrane patches on SMA treatment.** (**A**) SMA-resistant high-expression RC-LH1-X membranes have a low lipid:protein ratio and limited regions of the lipid bilayer. (**B**) Fusion with lipids or SMA-amenable bilayer-rich membranes introduces lipid-rich regions (pale green) between domains of closely packed RC-LH1-X complexes. (**C**) The addition of SMA causes solubilization of bilayer-rich regions as SMA-lipid nanodiscs (red/olive green). (**D**) This treatment liberates protein-rich membrane fragments that are sufficiently small to stay in solution (blue) during clearing ultracentrifugation spins and pass through the matrices of chromatography columns. (**E**) Extraction efficiencies of RC-LH1-PufX (blue), LH2 (green), and the cyt*bc*_1_ complex (magenta) in membranes prepared from cells grown under semi-aerobic (SA) or low light (LL) and medium light (ML) photosynthetic conditions. The left panel shows solubilization in 2.5% *w*/*v* SMA polymer, the center panel shows low light membranes solubilized in 3% *w*/*v* β-DDM, and the right panel shows results where no solubilizing agents were added. Solid bars show values obtained by spectroscopy, and hatched bars show values for cyt*c*_1_ by heme staining. Error bars indicate the standard error of the mean for three replicates. The inefficient solubilization was due to the polymer’s inability to disrupt the highly ordered and closely packed arrays formed by RC-LH1-PufX complexes. Figure 4A–D and caption are adopted from reference [118]. Figure 4E and caption are adopted from reference [114] with copyright permission.

**Figure 5 biomolecules-12-01076-f005:**
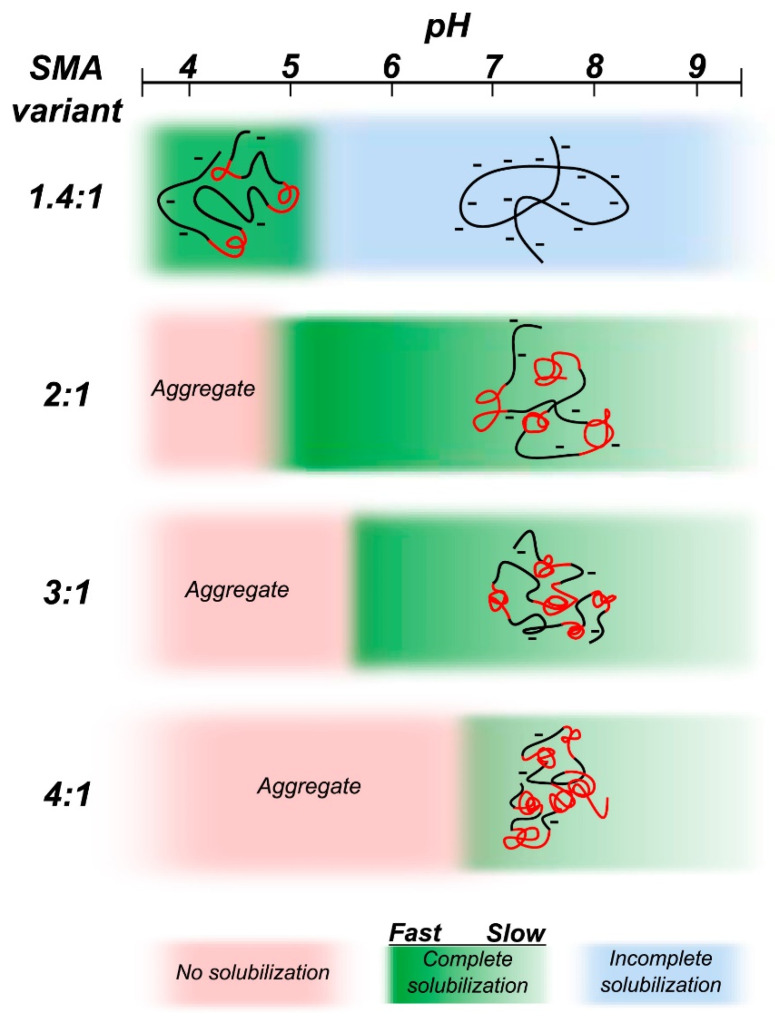
Schematic diagram that summarizes the effects of SMA composition and pH on the molecular conformation and solubilization efficiency of the SMA copolymer. The amphipathic polymer is represented as a cartoon in which the hydrophobic domains enriched in styrene units are shown in red, while the maleic acid-rich hydrophilic part of the polymer is shown in black. The efficiency of cell membrane solubilization is depicted according to color coding. (Dark green) Complete and fast solubilization; (blue) solubilization is induced but remains incomplete; and (red) the polymer is not able to solubilize at all due to self-assembly and aggregation. It should be noted that the exact conditions vary with the protein under investigation. This Figure and caption are adopted from reference [120] with copyright permission.

**Figure 7 biomolecules-12-01076-f007:**
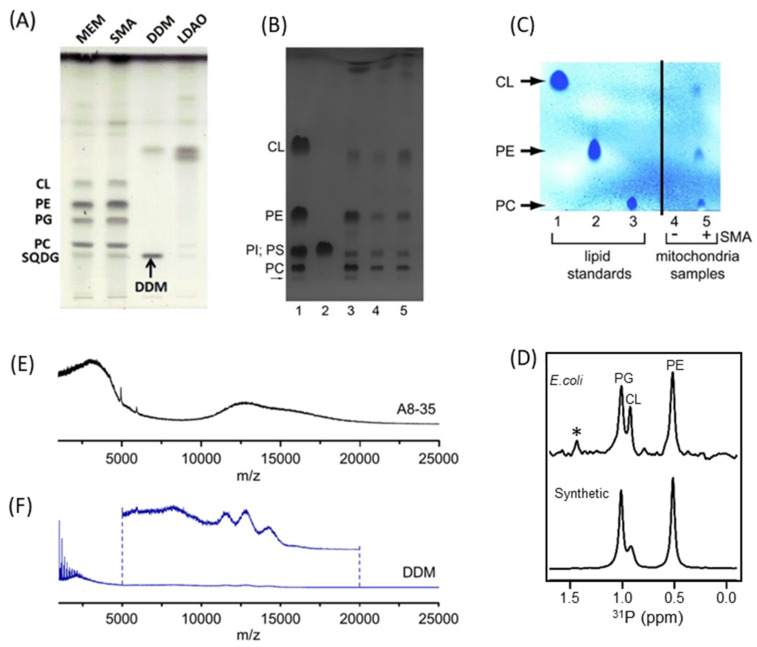
**Analysis of phospholipids in the local lipid polymer-nanodiscs.** (**A**) TLC of lipids present in membranes and nanodiscs was identified by running pure synthetic samples of each as a standard (not shown). Bands above the labeled lipids are attributed to photoreaction center ([RC] from the purple bacterium *Rhodobacter sphaeroides*) pigments. DDM was visualized, but LDAO did not stain. Additional bands in the DDM and LDAO profiles are unidentified. No lipids could be detected in the samples solubilized by detergents DDM and LDAO, indicating the detergent removal of lipids [133]. (**B**) TLC of chloroform-methanol extracts of the yeast mitochondria and of CytcO-SMA native nanodiscs. From left to right: the 1st and the 2nd lanes are lipid standards (0.04 mg CL, 0.05 mg of each of DOPC, DOPE, PI, and PS); the 3rd lane is the extract of the yeast mitochondria (the loaded lipids were extracted from a sample that originally contained ~0.6 mg protein); the 4th and 5th lanes are extracts of two preparations of CytcO-SMA. The loaded lipids (at the arrow) were extracted from a sample that originally contained ~0.2 mg protein. The bands in lane 4 are slightly weaker than those in lane 5, presumably because slightly less material was applied in the former. The bands were visualized by iodine staining [113]. (**C**) Mitochondria subjected to mock SMA incubation (lane 4), or incubated with SMA, to form mitochondrial-LipodisqsW (lane 5) were analyzed for lipid content by thin-layer chromatography. Lipid standards are shown in lanes 1–3 [130]. (**D**) ^31^P NMR spectra of native *E. coli* lipids present in the polymer-nanodiscs and of the synthetic lipids (a reference sample). PE: phosphatidylethanolamine, CL: cardiolipin, and PG: phosphatidylglycerol, * indicates the uncharacterized *E. coli* lipids [55]. (**E**,**F**) Mass spectrometry analysis of SMA extracted AcrB after exchange into A8–35 and DDM. Initial native MS results of A8–35_Ex (**E**) and DDM_Ex (**F**) [134]. Figure 7A,B and captions are adopted from the references [113,133] with copyright permission. Figure 7C–F and captions are adopted from the references [55,130,134].

**Figure 8 biomolecules-12-01076-f008:**
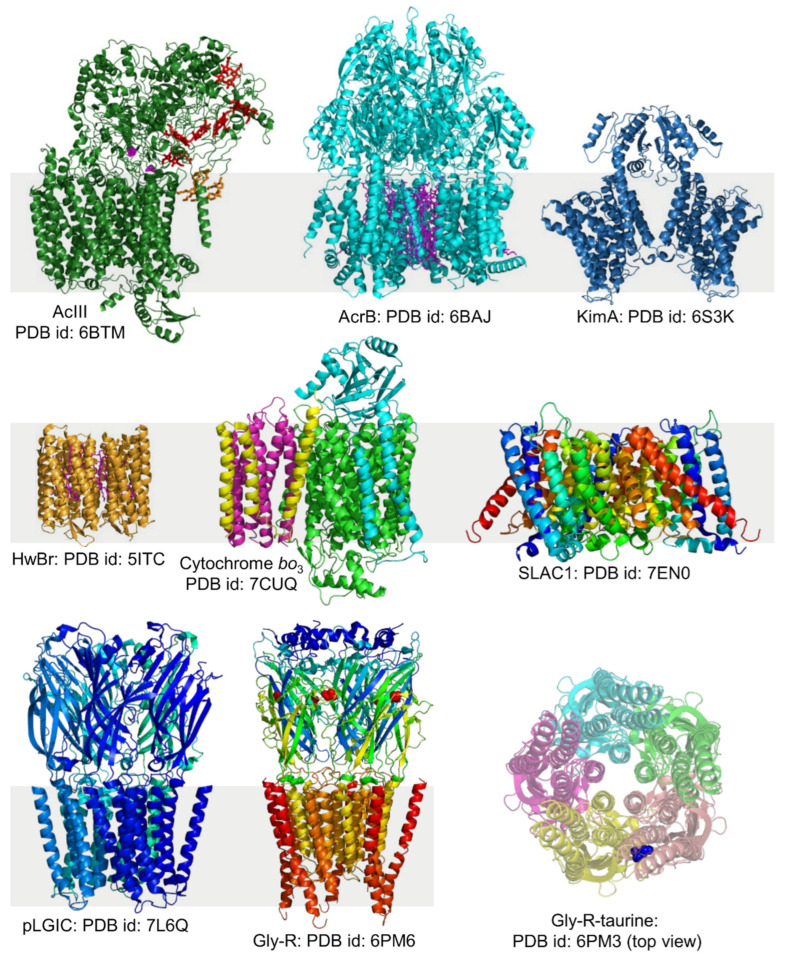
The structure of alternative Complex III (PDB id: 6BTM) from *Flavobacterium johnsoniae* [184], AcrB (PDB id: 6BAJ) from *E. coli* (K-12) [182], KimA (PDB id: 6S3K) from *Bacillus subtilis* [157], HwBr (PDB id: 5ITC) from *Haloquadratum walsbyi* [119], cytochrome bo3 (PDB id: 7CUQ) from *E. coli* [171], SLAC1 (PDB id: 7EN0) from *Brachypodium distachyon* SLAC1 [173], bacterial pLGIC (PDB id: 7L6Q) [172], the glycine receptor open conformation (PDB id: 6PM6; the Gly residues are shown in red spheres) and taurine-bound closed conformation (top view) (PDB id: 6PM3; taurine is shown in blue spheres) from zebrafish [185]. The 6PM3 structure is shown with 50% transparency to highlight taurine. ASIC1 (PDB id: 6VTK) from chicken [186], and EspP-BamA complex structure (PDB id: 7TTC) from *E. coli* [136]. The structures were generated using PyMOL.

**Figure 9 biomolecules-12-01076-f009:**
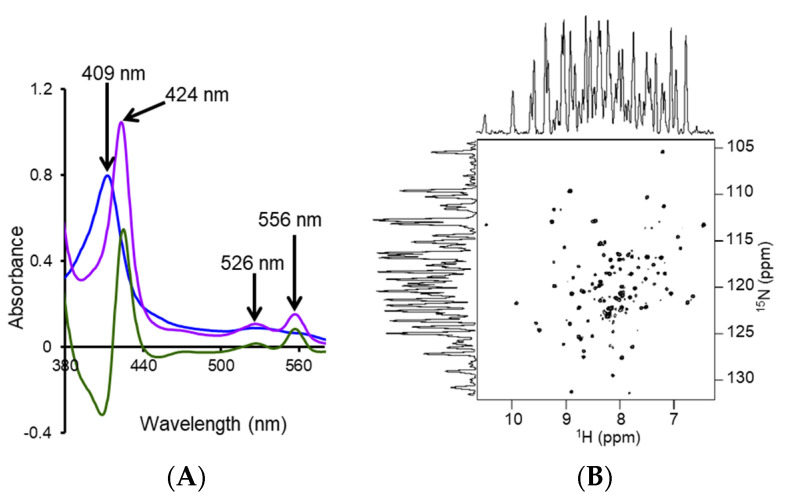
(**A**) Absorbance spectra of recombinant ~16-kDa rabbit cytochrome-b5 isolated in native *E. coli* native lipid-nanodiscs using an anionic SMA-EA polymer: oxidized form (409 nm) (blue), sodium dithionite-reduced form (424, 526 and 556 nm) (magenta), and difference spectra (reduced minus oxidized) (green). (**B**) 2D ^1^H/^15^N TROSY-HSQC NMR spectrum of ^15^N-labelled cytochrome-b5 in *E. coli* native lipid polymer-nanodiscs. This Figure and caption are adopted from reference [55].

**Figure 10 biomolecules-12-01076-f010:**
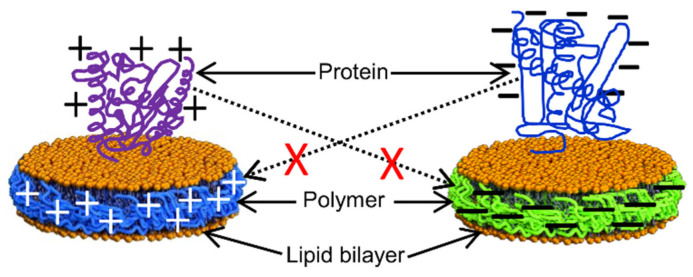
Schematic showing the lipid-nanodiscs containing positively-charged polymers and a positively-charged protein (**left**), negatively-charged polymers, and a negatively-charged protein (**right**). Due to opposite charges, non-specific interactions occur between the belt-forming polymers and the reconstituted protein at a given pH which would reduce the stability of nanodiscs and also lead to structural changes and aggregation. Hence, the synthetic polymer used and the membrane protein to be reconstituted/studied should possess the same net charge.

**Figure 11 biomolecules-12-01076-f011:**
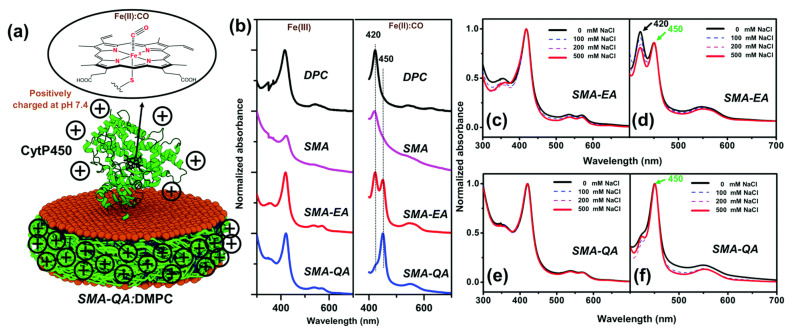
Reconstitution and functional characterization of CYP450 2B4 in differently charged polymer-nanodiscs and DPC micelles: (**a**) Schematic showing CYP450 with heme coordination spheres of the CO-bound state reconstituted in an SMA-QA:DMPC nanodisc. (**b**) UV-vis absorption spectra of CYP450 reconstituted in different SMA polymer-nanodiscs or in DPC micelles in its ferric state (left column) and in a ferrous state in complex with CO (right column). UV-vis absorption spectra of a positively-charged CYP450 reconstituted in negatively-charged SMA-EA nanodiscs: (**c**) in the presence of the indicated NaCl concentrations and (**d**) the ferrous-CO complex (**d**). UV-vis spectra of a positively-charged CYP450 reconstituted in positively-charged SMA-QA-based DMPC nanodiscs and (**e**) in the presence of NaCl and (**f**) the ferrous-CO complex. These results demonstrate the importance of a membrane mimetic and polymer charge in nanodisc for the functional reconstitution of membrane proteins. The inactive CYP450 (i.e., P420) in the presence of DPC detergent or an anionic-polymer (like SMA or SMA-EA) is undesirable. The positively-charged SMA-QA retains the functional form of CYP450. This Figure and caption are adopted from reference [199].

**Figure 12 biomolecules-12-01076-f012:**
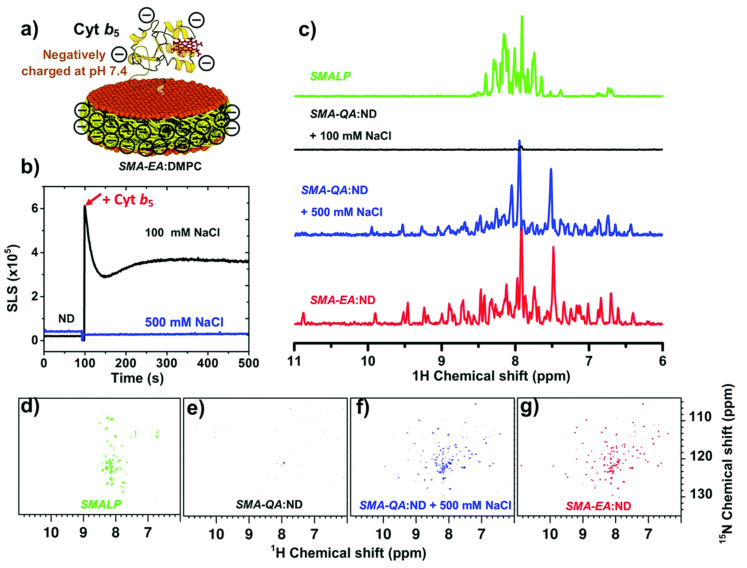
Reconstitution and structural characterization of cytochrome-b5 in various SMA-based DMPC-nanodiscs: (**a**) schematic representation of a negatively charged ~16 kDa rabbit cytochrome-b5 reconstituted in negatively charged SMA-EA-based DMPC-nanodiscs. (**b**) Static light scattering (SLS) profiles of cationic SMA-QA-based DMPC-nanodiscs containing cytochrome-b5 at low (100 mM) and high (500 mM) NaCl concentrations. (**c**) Projections of 2D ^1^H/^15^N TROSY-HSQC NMR spectra of a uniformly-^15^N-labelled cytochrome-b5 reconstituted in SMALP (**d**), SMA-QA with 100 mM NaCl (**e**), SMA-QA with 500 mM NaCl (**f**), and SMA-EA (**g**) DMPC-nanodiscs. The presence of aggregates in the sample containing 100 mM NaCl, indicated by the SLS profile in (**b**), explains the reason for the absence of resonances in the 2D NMR spectrum (**e**). On the other hand, the appearance of NMR resonance in (**f**) (and the SLS profile) due to the use of a high concentration of NaCl confirms the formation of non-specific charge-charge coulombic interactions between the positively-charged SMA-QA polymer belt and the negatively-charged cytochrome-b5. Although the use of high salt concentration enables NMR data acquisition, it is not physiologically relevant. It can damage proteins in NMR samples by causing serious radio-frequency-induced heating in the sample to. This Figure and caption are adopted from reference [199].

**Figure 13 biomolecules-12-01076-f013:**
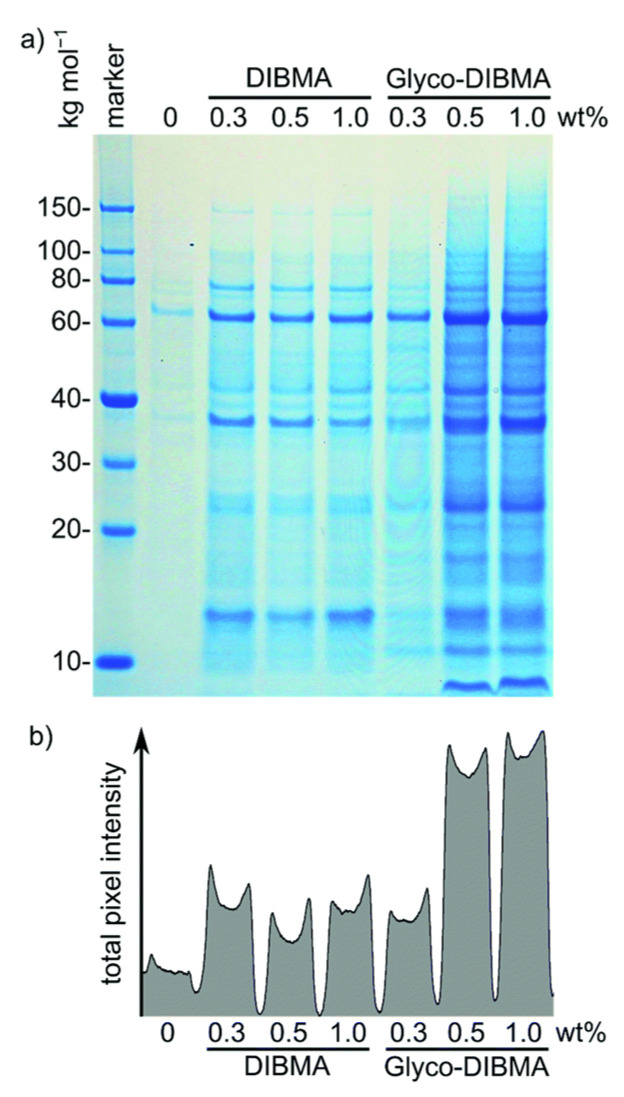
Isolation of the membrane proteome of *E. coli* cells into polymer-nanodiscs using Glyco-DIBMA and DIBMA polymers. Shown are (**a**) a Coomassie-stained gel after SDS-PAGE of polymer-solubilized membrane fractions and (**b**) a projection of the total pixel intensity across all lanes in the SDS-PAGE gel. Insoluble cell debris and water-soluble proteins were removed by centrifugation, and samples were gently agitated overnight at 23 °C in the presence of Glyco-DIBMA or DIBMA. Prior to SDS-PAGE, insoluble material and polymer were removed by ultracentrifugation and organic solvent extraction, respectively. A control without polymers was produced under otherwise identical conditions. This Figure and caption are adopted from reference [107].

**Figure 15 biomolecules-12-01076-f015:**
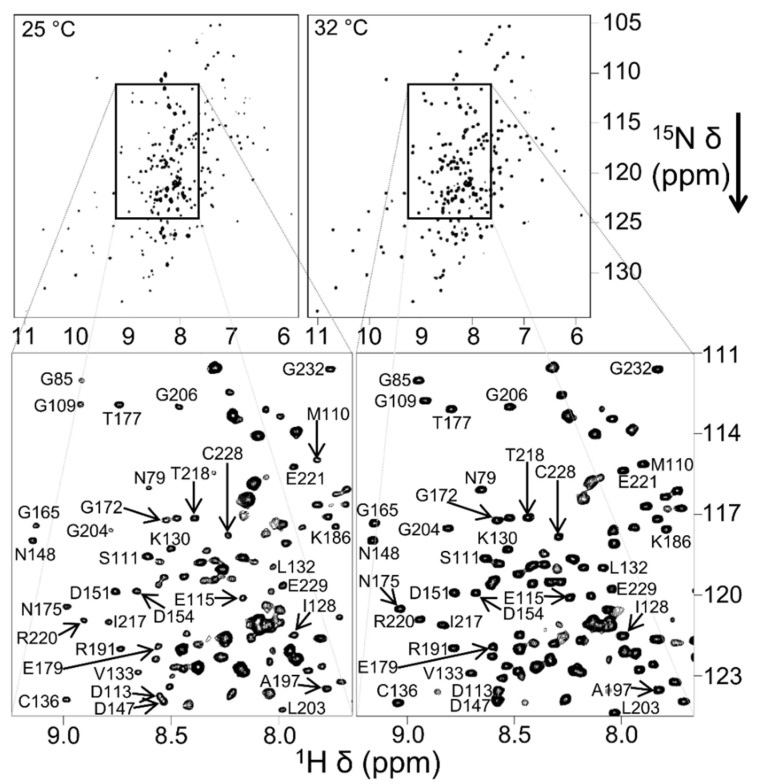
2D [^1^H–^15^N]-TROSY-HSQC NMR spectra of 75 μM ^15^N-labelled FBD in nanodiscs recorded at an 800 MHz NMR spectrometer. For easy reading, expanded regions are shown below, highlighting a few peaks with a substantial signal improvement at higher temperature (32 °C). The observation of well dispersed NMR spectral lines demonstrate the absence of any interaction between the polymer belt and FBD. Figure and caption are adopted from reference [58].

**Figure 16 biomolecules-12-01076-f016:**
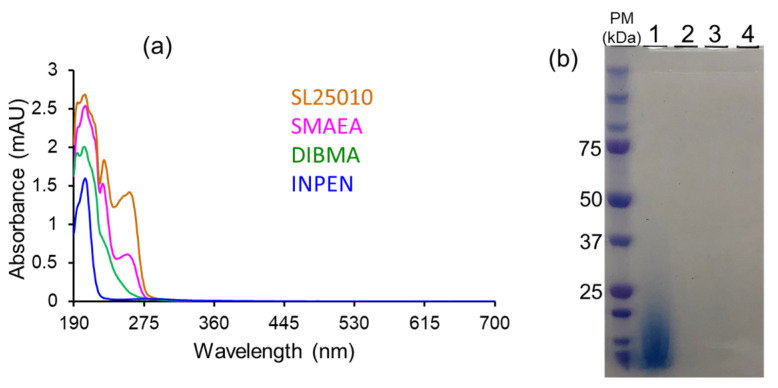
(**a**) UV-visible absorbance spectra of pentyl-inulin, DIBMA, SMAEA, and SL25010 polymers. Pentyl-inulin showed no absorbance in the 230–700 nm wavelength range. Due to aromatic rings, both SMA-based polymers SMA-EA and SL25010 showed substantial absorbance near 260 nm wavelength. (**b**) SDS-PAGE analysis of (lane-1) ~10 kDa SMA25010, (lane-2) ~2 kDa SMA-EA, (lane-3) ~12 kDa DIBMA, and (lane-4) ~3 kDa pentyl-inulin. This Figure and caption are adopted from reference [58].

**Table 1 biomolecules-12-01076-t001:** A list of membrane solubilizing detergents commonly used for membrane protein isolation.

**(1) Ionic Detergents**
Sodium dodecyl sulfate (SDS; anionic)
Deoxycholate (anionic; bile acid salt)
Sodium cholate (anionic; bile acid salt)
Calixarene (anionic)
N-lauryl sarcosine or sarkosyl (anionic)
Cetyltrimethylammonium bromide (CTAB; cationic)
Hexadecyltrimethylammonium bromide (cationic)
**(2) Zwitterionic detergents**
3-[(3-cholamidopropyl)dimethylammonio]-1-propanesulfonate (CHAPS)
3-[(3-cholamidopropyl)dimethylammonio]-2-hydroxy-1-propanesulfonate (CHAPSO)
n-dodecyl-N, N-dimethylamine-N-oxide (LDAO)
n-dodecyl phosphocholine (DPC)
**(3) Nonionic detergents ^a^**
Poly-oxyethyleneglycol lauryl ether
n-dodecyl-β-D-maltoside (DDM)
n-nonyl-β-D-glucoside (NG)
n-octylglucoside (OG)
Polyethylene glycol tert-octyl phenyl ether (Triton X-100)
Undecyl-β-D-maltoside (UDM)
Digitonin
Maltose neopentyl glycol (MNG)
Hecameg [6-*O*-(*N*-heptylcarbamoyl)-methyl-α-D-glucopyranoside (HG)

^a^ Long-chain (C_12_–C_14_) nonionic detergents are milder than short-chain (C_7_–C_10_) nonionic detergents.

**Table 2 biomolecules-12-01076-t002:** List of membrane proteins reconstituted in nanodiscs and studied by NMR spectroscopy.

Protein(s) Reconstituted	Nanodisc Type	Ref.
Human cytochrome P450 3A4 (CYP3A4)	MSP	[28]
Human voltage-dependent anion channel-1 (VDAC-1)	MSP	[29,30,31,32]
Human voltage-dependent anion channel-2 (VDAC-2)	MSP	[33]
VDAC N-terminal segment (NTS)	MSP	[34]
The transmembrane domain of stromal interaction molecule (STIM1-TM)	MSP	[35]
Bacterial β-barrel assembly machinery-A (BamA)	MSP	[31,36,37]
Bacteriorhodopsin	MSP	[38,39,40]
Outer membrane protein X (OmpX)	MSP	[30,31,32,38,40,41,42]
*α*–helical BLT2 G protein-coupled receptor	MSP	[41]
NTS_8–13_–NTSR1–Gα_i1_β_1_γ_1_ complex	MSP	[43]
hIAPP	MSP	[44]
Anti-apoptotic protein BclxL	MSP	[45]
Inner mitochondrial MPV17	MSP	[46]
Bak transmembrane helix	MSP	[47]
*Y. pestis* Omp adhesion invasion locus (Ail)	MSP	[48]
Human interleukin-8 (IL-8)-CXCR1(1–38) complex	MSP	[49]
Rabbit cytochrome-b5 + horse cytochrome C	4F peptide	[50]
Rabbit CYP450 2B4, rat CYP450 reductase FMN-binding domain, and rabbit cytochrome-b5	4F peptide	[51,52,53]
Cytochrome-b5 + CYP450	22A peptide	[54]
Pf1, p7 from human hepatitis C virus and human chemokine receptor CXCR1 (GPCR)	18A peptide	[21]
Rabbit cytochrome-b5	SMA-EA	[55]
Pf1 coat protein	SMA	[56,57]
MerFt, CXCR1 and Ail	SMA	[56]
Rat CYP450 reductase FMN-binding domain	Pentyl-inulin	[58]

**Table 3 biomolecules-12-01076-t003:** List of synthetic amphipathic polymers used for detergent-free isolation of membrane proteins directly from the cell membrane.

**(1) Ionic Polymers**
Styrene maleic acid copolymer (SMA) (1:1, 2:1, 3:1, 2.3:1, 1.2:1)
SMA-QA and SMA-EA
poly(styrene-*co*-maleimide) (SMI)
Diisobutylene maleic acid co-polymer (DIBMA)
**(2) Zwitterionic polymers**
zSMA, SMA-ED and SMA-Neut
**(3) Nonionic polymers**
Inulin functionalized with different hydrophobic moieties

## Data Availability

Not applicable.

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
