# Peer review of "Detergent-Free Isolation of Membrane Proteins and Strategies to Study Them in a Near-Native Membrane Environment"

_biomolecules, 2022, doi:10.3390/biom12081076_

Round 1

Reviewer 1 Report

Comments are attached.

Author Response

Reviewer #1

This review article entitled “Detergent-free isolation of membrane proteins and strategies to study them in a near-native membrane environment” is well describing that the ability of synthetic amphipathic polymers to isolate membrane proteins directly from the cell membrane, along with the associated membrane components such as lipids in a native-membrane, without the use of a detergent, has opened new avenues to study the structure and functions of membrane proteins using a variety of biophysical and biological approaches. This review is informative and covering the recent developments on the direct isolation and functional reconstitution of membrane proteins to form nonodiscs for structural biology studies and also high-resolution structure investigations of variety of membrane proteins are describing in detail. This reviewer, therefore, think that this review article is published to biomolecules in the present form.

Author’s Response:

We thank the Reviewer for careful reading of the manuscript, for positive comments, and for recommending the review for publication.

Reviewer 2 Report

In this manuscript, the authors review the recent strategies for isolation of membrane proteins without using detergents and their study in near-native membrane environment. These strategies mainly involve the formation of lipid or polymer nanodiscs. This is definitely a hot topic, since there are more and more evidence that lipids are required for both the stability and function of membrane proteins, and that even mild detergents can yield proteins in non native or non functional states.

Overall the review is understandable by both experts and non experts from the field. The figures are very informative. About 60% of the referenced articles are from the last 5 years. The challenges, advantages and limitations of these new strategies are well discussed. I have, however, one main criticism concerning the organization of the review which is something not clear for me and makes the review difficult to read. For instance, in the core paragraph 4, several different aspects are discussed: 1) the different polymers that have been used, 2) the procedure of formation of the nanodiscs, 3) examples of membrane proteins that have been isolated, 4) structural characterization of the isolated membrane proteins and 5) functional studies. For now, everything is mixed. I believe, that it will help the reader and make easier to find the important information if paragraph 4 is divided into several subparts where only one aspect is discussed in each subpart.

I also suggest the authors to read again carefully the manuscript and correct some clumsy sentences or typo, such as:

- line 36: "have reported many of the reported structures"

- line 600: cytoichrome b5

- line 620: those that function/stable in acid environments

- line 672-673: SMA-EA and SMA-QA, due to the high charge density, the use of the polymers,... I believe that it is more simple to say: the use of SMA-EA and SMA-QA, due to their high charge density,....

- line 676 : chare instead of charge

Author Response

Reviewer #2

Comments and Suggestions for Authors

In this manuscript, the authors review the recent strategies for isolation of membrane proteins without using detergents and their study in near-native membrane environment. These strategies mainly involve the formation of lipid or polymer nanodiscs. This is definitely a hot topic, since there are more and more evidence that lipids are required for both the stability and function of membrane proteins, and that even mild detergents can yield proteins in non native or non functional states.

Overall the review is understandable by both experts and non experts from the field. The figures are very informative. About 60% of the referenced articles are from the last 5 years. The challenges, advantages and limitations of these new strategies are well discussed. I have, however, one main criticism concerning the organization of the review which is something not clear for me and makes the review difficult to read. For instance, in the core paragraph 4, several different aspects are discussed: 1) the different polymers that have been used, 2) the procedure of formation of the nanodiscs, 3) examples of membrane proteins that have been isolated, 4) structural characterization of the isolated membrane proteins and 5) functional studies. For now, everything is mixed. I believe, that it will help the reader and make easier to find the important information if paragraph 4 is divided into several subparts where only one aspect is discussed in each subpart.

I also suggest the authors to read again carefully the manuscript and correct some clumsy sentences or typo, such as:

Response to the comments

As per the reviewer’s suggestion, we have revised Section 4 by dividing it into sub-sections as shown in the manuscript file with highlights of changes made in the revision.

1) line 36: "have reported many of the reported structures"

Corrected.

2) line 600: cytoichrome b5

Typo is corrected.

3) line 620: those that function/stable in acid environments

Corrected.

4) line 672-673: SMA-EA and SMA-QA, due to the high charge density, the use of the polymers,... I believe that it is more simple to say: the use of SMA-EA and SMA-QA, due to their high charge density,....

Simplified the sentence as per the reviewer’s suggestion

5) line 676 : chare instead of charge

Typo is Corrected.

We have also added three additional Figures (4, 7, and 9) to make the review more readable and informative. Figure 8 is updated with the ASIC1 structure. With these changes, we believe that the manuscript is much improved and comprehensively covers the published studies in the field.

Round 2

Reviewer 1 Report

This review is well covering the recent developments on the isolation and functional reconstitution of membrane proteins to form lipid/polymer nanodiscs for structural biology studies and also high-resolution structural investigations of variety of membrane proteins are describing in detail. Paragraph 4 is well revised according to the reviewers comments and make the review more readable and informative. This reviewer, therefore, think that this review article is published to Biomolecules in present form.

Reviewer 2 Report

I thank the authors to have taken my suggestions into consideration. I believe that the manuscript is now suitable for publication in Biomolecules.